# The Lon protease temporally restricts polar cell differentiation events during the *Caulobacter* cell cycle

**Deike J Omnus[†], Matthias J Fink[†], Klaudia Szwedo, Kristina Jonas***

Science for Life Laboratory and Department of Molecular Biosciences, The Wenner-Gren Institute, Stockholm University, Stockholm, Sweden

**Abstract** The highly conserved protease Lon has important regulatory and protein quality control functions in cells from the three domains of life. Despite many years of research on Lon, only a few specific protein substrates are known in most organisms. Here, we used a quantitative proteomics approach to identify novel substrates of Lon in the dimorphic bacterium *Caulobacter crescentus*. We focused our study on proteins involved in polar cell differentiation and investigated the developmental regulator StaR and the flagella hook length regulator FliK as specific Lon substrates in detail. We show that Lon recognizes these proteins at their C-termini, and that Lon-dependent degradation ensures their temporally restricted accumulation in the cell cycle phase when their function is needed. Disruption of this precise temporal regulation of StaR and FliK levels in a Δ*lon* mutant contributes to defects in stalk biogenesis and motility, respectively, revealing a critical role of Lon in coordinating developmental processes with cell cycle progression. Our work underscores the importance of Lon in the regulation of complex temporally controlled processes by adjusting the concentrations of critical regulatory proteins. Furthermore, this study includes the first characterization of FliK in *C. crescentus* and uncovers a dual role of the C-terminal amino acids of FliK in protein function and degradation.

**\*For correspondence:**
kristina.jonas@su.se

[†]These authors contributed equally to this work

**Competing interest:** The authors declare that no competing interests exist.

## Introduction

Intracellular proteolysis is a critical process in all cell types that is carried out by dedicated proteases. By removing damaged and non-functional proteins, proteases are necessary for maintaining protein homeostasis, in particular under stress conditions that threaten the proteome. Additionally, proteases have important regulatory roles in precisely adjusting the amounts of specific functional proteins, thus complementing transcriptional and post-transcriptional control mechanisms. Because of their important cellular functions, human proteases are considered as promising therapeutic targets (***Bota and Davies, 2016***) and their bacterial counterparts as potential antimicrobial drug targets (***Culp and Wright, 2016***). Hence, extending the knowledge of the substrate pools of specific proteases and the mechanisms underlying substrate selection is vital.

In prokaryotes and in the mitochondria and chloroplasts of eukaryotes, the majority of proteins is degraded by ATP-dependent proteases of the AAA+ (ATPases associated with various cellular activities) protein family (***Sauer and Baker, 2011***). The protease Lon was the first ATP-dependent protease to be identified and is widely conserved across the three domains of life (***Gur, 2013***). Lon forms a hexameric protease complex, of which each monomer contains three functional domains: an N-terminal domain, an ATP-dependent unfoldase domain, and a peptidase domain forming the proteolytic chamber (***Gur, 2013***). As a heat shock protein (***Phillips et al., 1984***), Lon is upregulated in response to protein unfolding stress, such as thermal stress, and contributes to the degradation of unfolded and misfolded proteins that accumulate under these conditions (***Gur, 2013***; ***Gur and Sauer, 2008***; ***Tomoyasu et al., 2001***). In addition to its role in protein quality control, Lon exerts important

regulatory functions that can be traced to the degradation of specific sets of native substrate proteins involved in stress responses, metabolism, pathogenicity, and cell cycle progression (**Tsilibaris et al., 2006**).

Despite many years of research on Lon proteases, the number of validated Lon substrates is small in most organisms. Most knowledge about Lon has been obtained by studying bacterial Lon orthologs. In *Escherichia coli*, Lon specifically degrades several stress-induced regulators (**Griffith et al., 2004**; **Langklotz and Narberhaus, 2011**; **Mizusawa and Gottesman, 1983**), metabolic enzymes (**Arends et al., 2018**) as well as antitoxins of toxin-antitoxin systems (**Muthuramalingam et al., 2016**), and in several pathogenic bacteria, Lon was shown to degrade regulators of pathogenicity (**Joshi et al., 2020**; **Puri and Karzai, 2017**), thus playing important roles in the regulation of virulence pathways. In the alpha-proteobacterium *Caulobacter crescentus*, the number of identified Lon substrates to date is small, however, Lon is known to impact cell cycle progression by degrading essential cell cycle regulators.

The *C. crescentus* cell cycle is characterized by an asymmetric cell division event and morphologically distinct cell cycle phases (**Curtis and Brun, 2010**). Each division yields a flagellated and piliated swarmer cell and a sessile stalked cell. While the daughter stalked cell initiates DNA replication immediately after cell division, the daughter swarmer cell must differentiate into a stalked cell before entering S-phase. Faithful progression through the *C. crescentus* cell cycle relies on precise coordination of the polar differentiation events that trigger flagella, pili, and stalk biosynthesis with core cell cycle events, such as DNA replication and cell division (**Curtis and Brun, 2010**). Previous work established that around one-third of all genes in *C. crescentus* show cell cycle-dependent fluctuations in their expression levels (**Fang et al., 2013**; **Laub et al., 2000**). Many of the corresponding proteins have important developmental functions and peak in abundance in the cell cycle phase in which their function is most needed (**Laub et al., 2000**). In addition to transcriptional regulatory mechanisms, active proteolysis must occur to rapidly adjust protein concentrations to enforce these transcriptional changes (**Grünenfelder et al., 2001**). However, only a relatively small subset of cell cycle-regulated factors with developmental functions has so far been found to be subject to proteolysis in *C. crescentus* and the contributions of distinct proteases in this process remain incompletely defined.

Previous work established that the ClpP protease with its unfoldases ClpA and ClpX has key roles in *C. crescentus* development by mediating the temporally and spatially controlled degradation of several important cell cycle regulators (**Joshi and Chien, 2016**; **Schroeder and Jonas, 2021**). In addition to ClpXP, Lon plays an important role in *C. crescentus* cell cycle regulation (**Joshi and Chien, 2016**; **Schroeder and Jonas, 2021**). It degrades the swarmer cell-specific transcription factor SciP (**Gora et al., 2013**), the methyltransferase and transcriptional regulator CcrM (**Wright et al., 1996**), and the conserved replication initiator DnaA (**Jonas et al., 2013**). Lon-dependent degradation of SciP and CcrM contributes to their cell cycle-dependent regulation (**Gora et al., 2013**; **Wright et al., 1996**), while DnaA degradation by Lon ensures rapid clearance of the protein at the onset of nutritional and proteotoxic stress, preventing cell cycle progression under these conditions (**Felletti et al., 2019**; **Jonas et al., 2013**; **Leslie et al., 2015**). Although *C. crescentus* cells lacking Lon are viable, they grow more slowly and show aberrant chromosome content and division defects (**Leslie et al., 2015**; **Wright et al., 1996**), which can in part be attributed to the stabilization of DnaA, SciP, and CcrM. Interestingly, Δ*lon* cells exhibit also characteristic developmental defects, that is, elongated stalks and motility defects (**Wright et al., 1996**; **Yang et al., 2018**), suggesting that Lon degrades additional substrates involved in cell differentiation.

Here, using a quantitative proteomics approach, we identified several proteins involved in the dimorphic life cycle of *C. crescentus* as novel Lon substrates. We studied in detail the transcriptional regulator of stalk biogenesis, StaR, and the flagella hook length regulator FliK as specific Lon substrates. We show that Lon is required to establish cell cycle-dependent fluctuations of these regulatory proteins, thereby contributing to their precise temporal accumulation during the cell cycle phase in which their function is needed. Furthermore, we demonstrate that the increased abundance of these proteins results in aberrant stalk length and motility defects. Taken together, our work revealed a critical role of Lon in coordinating cell differentiation with core cell cycle events in *C. crescentus*.

## Results

### A quantitative proteomics approach identifies novel putative Lon substrates involved in *Caulobacter* development

Previous work established that Lon degrades the cell cycle regulators DnaA, CcrM, and SciP in *C. crescentus* (*Gora et al., 2013*; *Jonas et al., 2013*; *Wright et al., 1996*). Absence of Lon, either in a Δ*lon* strain or following Lon depletion, results in increased stability and abundance of these proteins (*Figure 1A–B*). Conversely, *lon* overexpression leads to lower protein abundance of DnaA, CcrM, and SciP (*Figure 1C*). Based on these results, we reasoned that it should be possible to identify novel Lon substrates by monitoring proteome-wide differences in protein stability and protein abundance in strains lacking or overexpressing *lon* in comparison to wild-type (WT) cells. Thus, we sampled cells from the following strain backgrounds and conditions for quantitative proteomics analysis: (1) 0, 15, and 30 min following protein synthesis shut down in the WT to assess protein stability in the presence of Lon; (2) 0, 15, and 30 min following protein synthesis shut down in Δ*lon* cells to examine protein stability in the absence of Lon; (3) before and after 4.5 hr of Lon depletion in a $P_{van}$-dependent Lon depletion strain; and (4) before and after 1 hr of inducing *lon* overexpression from a medium-copy plasmid. Tandem mass tag (TMT) labeling and mass spectrometrical (MS) analysis were used to detect proteome-wide differences in protein levels across these samples. We obtained signals for 2270 or 2261 proteins, respectively, in our two biological replicates and sorted the proteins with respect to four criteria (see Materials and methods for details): (1) to be more stable in Δ*lon* cells than in the WT after 30 min of translation inhibition, (2) to be present in higher abundance at t=0 in Δ*lon* cells compared to the WT, (3) to be upregulated after Lon depletion compared to non-depleting conditions, and (4) to be downregulated following xylose-induced *lon* overexpression compared to non-inducing conditions (*Figure 1D*). We identified 26 proteins that fulfilled all four criteria, one of them being DnaA. 120 proteins fulfilled three criteria, and because CcrM and SciP were in this group of proteins (*Figure 1D*), we considered proteins satisfying either three or four criteria as putative Lon substrates. The 146 proteins that fell into this group contained proteins from all major functional categories (*Figure 1E*). In comparison to all detected proteins, the group of putative Lon substrates contained a lower proportion of metabolic proteins and proteins involved in core genetic information processing, whereas proteins involved in cell cycle and cell differentiation processes, signal transduction and stress responses as well as unclassified proteins showed a higher relative abundance (*Figure 1E*). Notably, we did not detect FixT and HipB2 in our proteomics experiment, two other recently reported Lon substrates in *C. crescentus* (*Stein et al., 2020*; *Zhou et al., 2021*).

Since Δ*lon* cells have previously been shown to exhibit developmental defects (*Wright et al., 1996*; *Yang et al., 2018*), we focused this study on the group of potential Lon substrates annotated to have functions in cell cycle and cell differentiation processes. This group included proteins involved in central cell cycle regulation, chromosome partitioning, stalk morphogenesis as well as motility and chemotaxis (*Figure 1F*). Interestingly, according to previously published RNA-sequencing data, a large subset of the genes encoding these proteins are subject to cell cycle-dependent regulation and peak in their expression during a specific cell cycle phase (*Figure 1F*; *Lasker et al., 2016*; *Schrader et al., 2016*).

### The developmental regulator StaR is a specific Lon substrate

One of the proteins that satisfied the four criteria in our proteomics experiment most clearly was the transcriptional regulator StaR, a protein previously reported to be involved in the regulation of stalk biogenesis and holdfast production (*Figure 1D and F*; *Biondi et al., 2006*; *Fiebig et al., 2014*). Like the three known substrates DnaA, CcrM, and SciP, StaR was more stable in the Δ*lon* strain compared to the WT and showed increased steady-state levels in Δ*lon* and Lon-depleted cells, but reduced protein levels in cells overexpressing *lon* (*Figure 2A*). To validate these proteomics data, we monitored the stability of StaR in WT and Δ*lon* cells using a StaR-specific antibody (*Fiebig et al., 2014*). In the WT, the levels of StaR were below the limit of detection, however, we observed a strong and stable band corresponding to StaR in the Δ*lon* strain (*Figure 2B*), indicating that StaR is upregulated in Δ*lon* cells, a result that is consistent with the proteomics data (*Figure 2A*). To analyze the rate of StaR degradation in the presence of Lon, we expressed *staR* from a medium-copy vector to elevate its levels. In this strain, StaR was degraded with a half-life of approximately 12 min, when Lon was present (*Figure 2C*). Absence of Lon resulted again in complete stabilization of StaR and increased

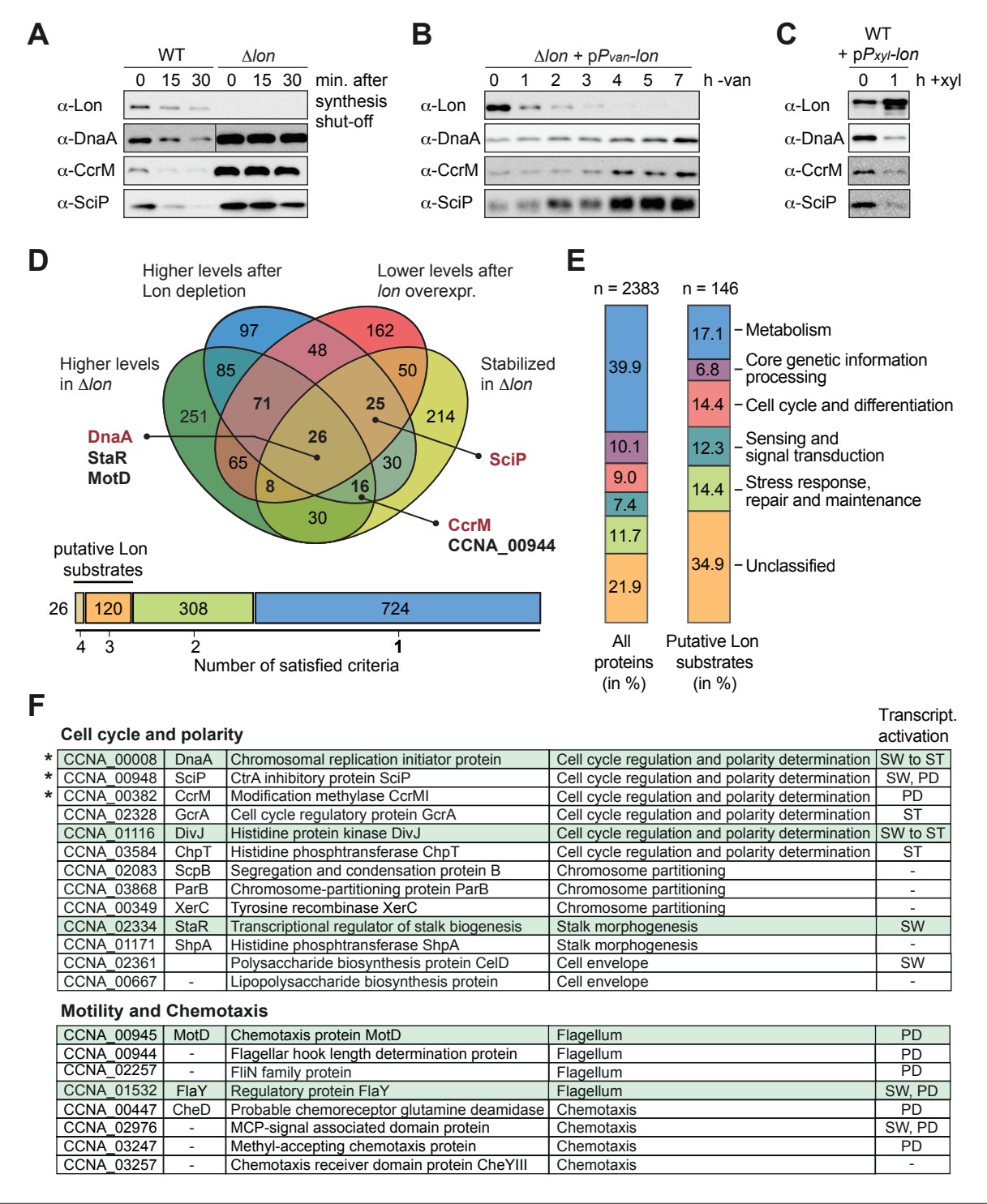

**Figure 1.** A quantitative proteomics approach identifies putative Lon substrates in *Caulobacter crescentus*. (**A**) In vivo stability assays of Lon, DnaA, CcrM, and SciP in the wild-type (WT) and the Δ*lon* strain (LS2382). Protein synthesis was shut down by addition of chloramphenicol at t=0 and remaining protein levels were measured after 15 and 30 min. The protein levels at t=0 correspond to the steady-state levels of these proteins in the WT and the Δ*lon* (LS2382) mutant. (**B**) Protein levels of Lon, DnaA, CcrM, and SciP over 7 hr of Lon depletion. The expression of *lon* was shut off by transferring the $P_{van}$-dependent Lon depletion strain (ML2022) from PYE supplemented with vanillate (van) to PYE lacking vanillate (−van). (**C**) Protein levels of Lon, DnaA, CcrM, and SciP in a $P_{xyl}$-dependent *lon* overexpression strain (ML2010) before (0) and 1 hr after induction with xylose (1 hr+xyl). (**D**) Venn chart showing groups of proteins that meet the following criteria and how they overlap: (1) to be present in higher abundance at steady state in Δ*lon* cells compared to the WT (green circle), (2) to be upregulated after 4.5 hr of Lon depletion compared to non-depleting conditions (blue circle), (3) to be downregulated

*Figure 1 continued*

after 1 hr of induced *lon* overexpression compared to non-inducing conditions (red circle), and (4) to be stabilized in Δ*lon* cells compared to the WT 30 min after translation shut off (yellow circle). Previously confirmed Lon substrates (shown in red) and the proteins investigated in this study (shown in black) are highlighted. The bar graph below the Venn chart indicates the number of proteins satisfying four, three, two, or one criteria. Proteins satisfying either three or four criteria were considered as putative Lon substrates. (**E**) Putative Lon substrates identified in (**D**) sorted by functional category (right bar graph). For comparison, the sorting of all detected proteins into functional categories is shown (left bar graph). (**F**) Tables listing the putative Lon substrates with functions in cell cycle and polarity as well as motility and chemotaxis. Previously known Lon substrates are marked with an asterisk. Proteins satisfying four criteria are highlighted in green. The cell cycle phase in which the expression of the listed proteins is transcriptionally induced is indicated (SW: swarmer cell, ST: stalked cell, PD: predivisional cell, SW to ST: swarmer to stalked cell transition).

The online version of this article includes the following figure supplement(s) for figure 1:

**Source data 1.** Proteomics data.

**Source data 2.** Unprocessed Western blot images.

steady-state levels, which is in line with our proteomics data, and confirms that StaR degradation depends on Lon. To directly test if StaR is a Lon substrate, we performed an in vitro degradation assay with purified StaR and Lon. This assay showed that Lon readily degrades StaR in an ATP-dependent manner (*Figure 2D*). Hence, StaR is a Lon substrate and no additional factors are required for recognizing and degrading StaR, at least in vitro.

We also investigated if Lon recognizes StaR via a degron sequence at one of the termini and monitored the degradation of FLAG-tagged StaR variants. Addition of the 3xFLAG tag to the C-terminus of StaR (StaR-F) completely abolished degradation (*Figure 2E*), indicating that a freely accessible C-terminus of StaR is required for degradation by Lon. Addition of the tag to the N-terminus of StaR (F-StaR) still enabled notable degradation within 60 min after shutting down protein synthesis (*Figure 2E*). However, degradation of this N-terminally tagged StaR was also observed in the Δ*lon* strain, suggesting that another protease degrades this StaR variant, probably because of changes in StaR folding that result from the addition of the tag. These data show that native N-termini and C-termini of the protein are required for Lon-dependent degradation.

## Lon ensures cell cycle-dependent regulation of StaR abundance

As a transcriptional regulator of stalk biogenesis and holdfast production, StaR function is expected to be particularly needed at the beginning of the cell cycle, when the swarmer cell differentiates into a stalked cell (*Figure 3A*). Indeed, previously published RNA sequencing data show that *staR* mRNA levels fluctuate during the cell cycle, peaking in the swarmer and early stalked cell before declining during S-phase and remaining low until cell division (*Figure 3B*; *Lasker et al., 2016*; *Schrader et al., 2016*). Furthermore, existing ribosome profiling data show that StaR is translated predominantly in the swarmer and early stalked cells, but only at low levels during the remaining cell cycle (*Lasker et al., 2016*; *Schrader et al., 2016*). Quantification of StaR protein levels by Western blot analysis in synchronized cultures showed that protein levels follow this expression pattern in WT cultures (*Figure 3C*). StaR was detectable within 30 min after synchronization before it was strongly downregulated and remained below the limit of detection for the rest of the cell cycle. Strikingly, the cell cycle-dependent changes in protein abundance were absent in the Δ*lon* strain, in which StaR levels remained high until 75 min after synchronization (*Figure 3C*). Thus, Lon-dependent degradation of StaR is required for establishing oscillations of StaR levels during the cell cycle.

## Lon-mediated StaR proteolysis is required for proper stalk biogenesis

Next, we assessed the importance of Lon-mediated StaR proteolysis for correct stalk biogenesis. Consistent with a previous study (*Wright et al., 1996*), we observed that the stalks of Δ*lon* cells are significantly elongated compared to the WT (*Figure 3D–E*). Because StaR overexpression was shown to lead to an increase of stalk length (*Biondi et al., 2006*), we reasoned that the abnormal stalk length of Δ*lon* cells might be caused by the higher abundance of StaR in these cells. To address this hypothesis, we introduced the Δ*staR* deletion into the Δ*lon* strain background and assessed stalk length of this Δ*staR* Δ*lon* double mutant (*Figure 3D–E*, *Figure 3—figure supplement 1*). The Δ*staR* Δ*lon* mutant phenocopied the Δ*staR* single mutant, in which stalks are shortened compared to Δ*lon* cells, demonstrating that Lon affects stalk length through StaR (*Figure 3D–E*). To further investigate the relationship between Lon, StaR, and stalk lengths, we also analyzed stalk morphology under

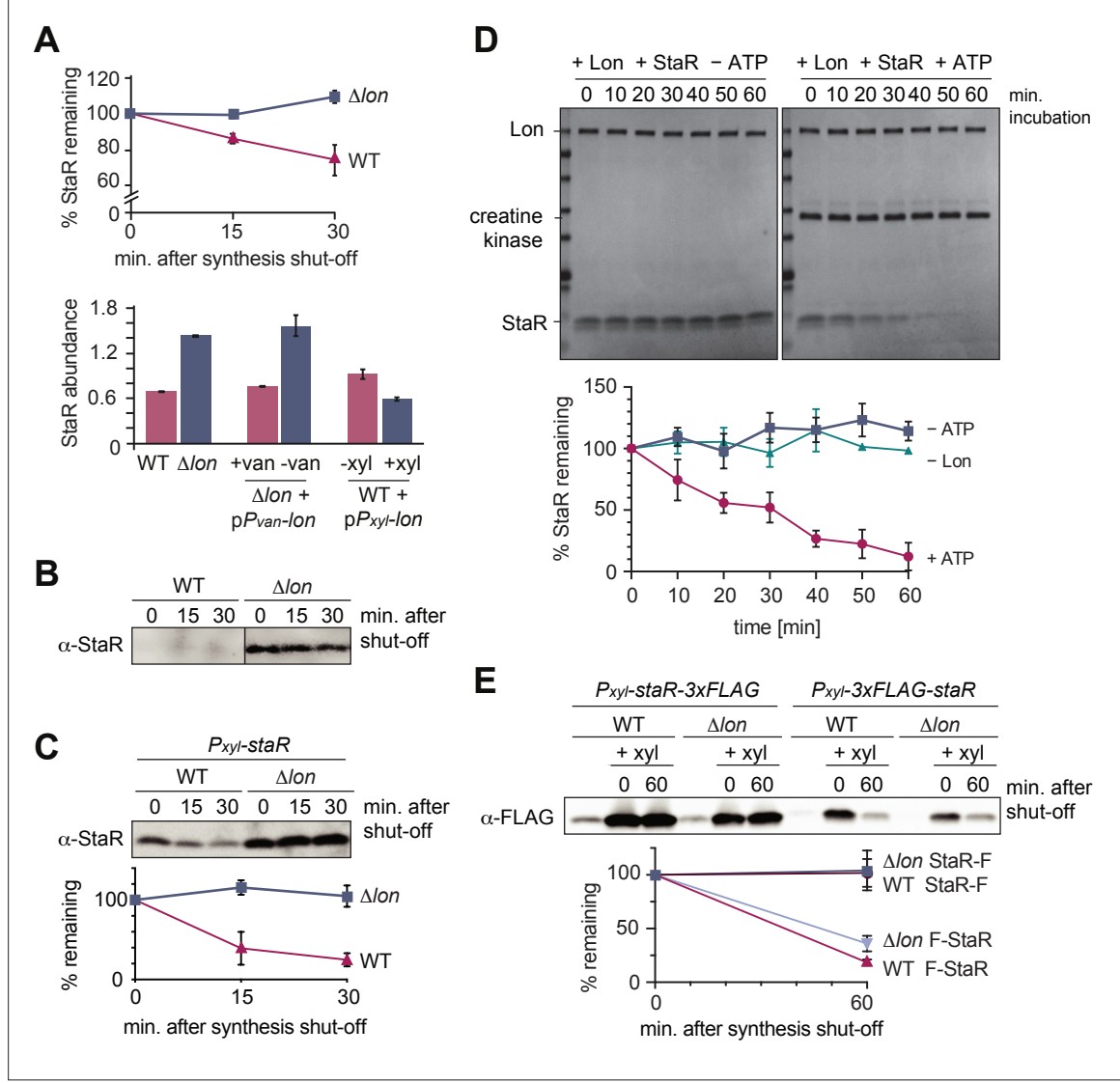

**Figure 2.** The developmental regulator StaR is a Lon substrate. (**A**) Proteomics data obtained for StaR. The upper graph shows StaR stability in wild-type (WT) and Δ*lon* (LS2382) cells and the lower graph StaR abundance in the different strain backgrounds and conditions as determined by mass spectrometry. Each data point represents the mean protein abundance of the two experimental replicates, error bars show standard deviations. (**B**) In vivo degradation assay of StaR at native expression levels in WT and Δ*lon* (KJ546) cells. (**C**) In vivo degradation assay of StaR after xylose-induced *staR* overexpression in WT and Δ*lon* (KJ546) backgrounds. The graph shows mean values and standard deviations of relative protein levels after protein synthesis shut off determined from three independent experiments. (**D**) In vitro assay showing degradation of StaR by Lon. 4 μM StaR and 0.125 μM Lon hexamer was incubated in the presence (+ATP) or absence (−ATP, control) of an ATP regeneration system. The graph shows the relative StaR levels normalized to Lon or CK (in case of the −Lon sample) levels of three independent experiments and are represented as means with standard deviations. (**E**) N-terminally and C-terminally tagged StaR (3xFLAG-StaR and StaR-3xFLAG, respectively) were expressed ectopically by xylose induction and their protein levels were assessed in WT and Δ*lon* (KJ546) cells prior to induction (first lanes of each set of three) and after induction either before (t=0 min) or after shutting off protein synthesis (t=60 min). The graph shows the mean values of relative protein levels and standard deviations of three independent experiments.

The online version of this article includes the following figure supplement(s) for figure 2:

**Source data 1.** Unprocessed Western blot and protein gel images.

phosphate-limiting conditions, in which stalks are drastically elongated in *C. crescentus* (*Schmidt and Stanier, 1966*). Stalk length in the different strain backgrounds followed the same trend as under optimal conditions (*Figure 3—figure supplement 1*). However, all four strains were able to strongly elongate their stalks under phosphate starvation, suggesting that StaR and Lon are not required

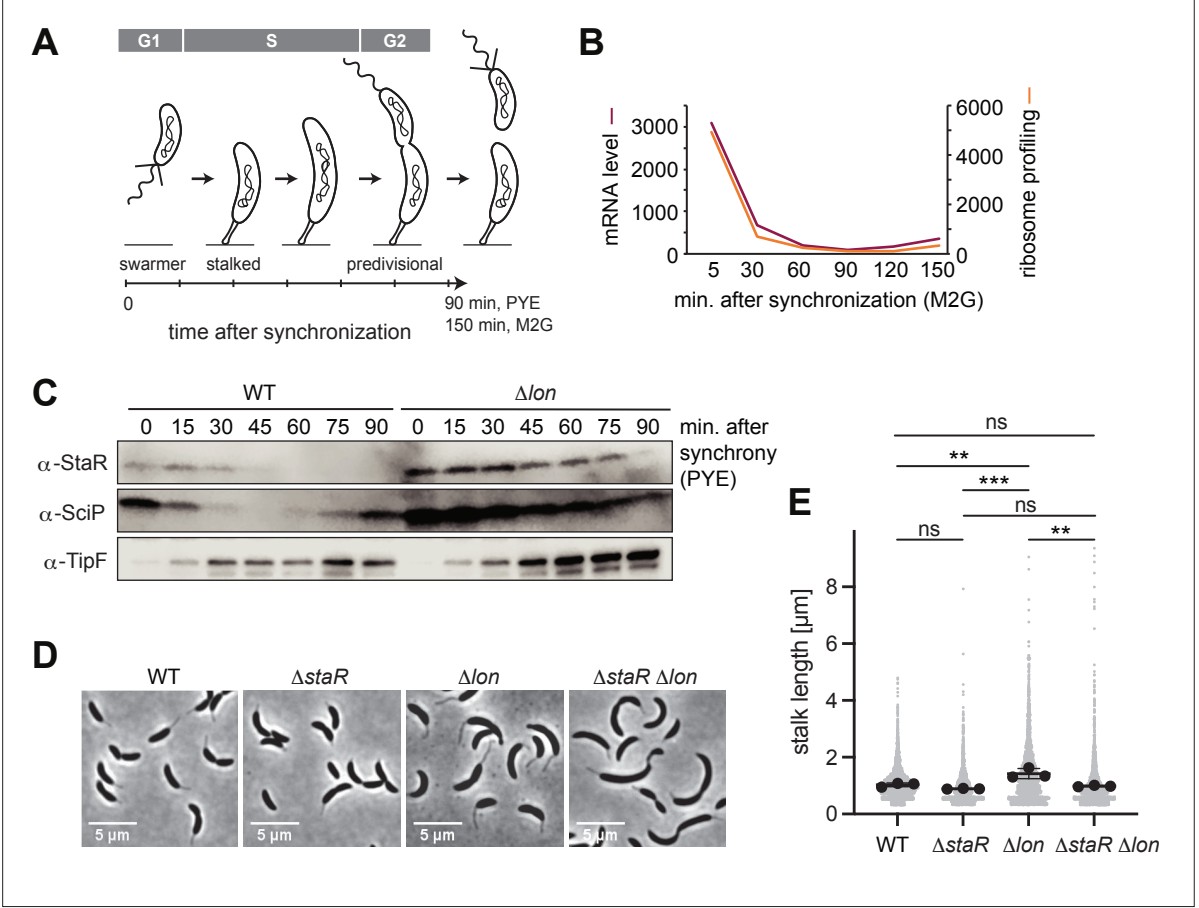

**Figure 3.** Lon ensures cell cycle-dependent accumulation of StaR and proper stalk length regulation. (**A**) Schematic illustration of the *Caulobacter crescentus* cell cycle and how the different cell cycle phases are associated with distinct morphological states. The time line indicates how the cell cycle progresses over time. (**B**) RNA-sequencing and ribosome profiling data for *staR*, as published previously (*Lasker et al., 2016*; *Schrader et al., 2016*). (**C**) Protein levels of StaR, SciP, and TipF in synchronized wild-type (WT) and Δ*lon* (KJ546) culture over 90 min following release of swarmer cells into PYE medium. Western blots for SciP and TipF were included as controls for proteins with well-characterized cell cycle patterns (*Davis et al., 2013*; *Gora et al., 2010*). (**D**) Phase contrast microscopy images depicting morphological differences between *C. crescentus* WT the single mutants Δ*staR* and Δ*lon* (KJ546) and the double mutant Δ*staR*Δ*lon* when grown in PYE at 30°C. (**E**) Quantifications of stalk length under optimal conditions (PYE, 30°C) of the strains shown in (**D**). N was at least 1800 total for each strain obtained from three biological replicates. Statistical significance was determined by ordinary one-way ANOVA (Tukey's multiple comparisons test: WT vs. Δ*lon* p=0.0047, **; Δ*staR* vs. Δ*lon* p=0.0007, ***; Δ*lon* vs. Δ*staR*Δ*lon* p=0.0025, **; WT vs. Δ*staR* p=0.3591, not significant; Δ*staR* vs. Δ*staR*Δ*lon* p=0.6398, not significant; WT vs. Δ*staR*Δ*lon* p=0.9447, not significant).

The online version of this article includes the following figure supplement(s) for figure 3:

**Source data 1.** Unprocessed Western blot images.

**Figure supplement 1.** Quantification of stalk length under standard (M2G) and phosphate-limiting conditions (M5G).

for starvation-dependent stalk elongation. Taken together, our data demonstrate that Lon-mediated degradation of StaR is required for proper stalk biogenesis during the cell cycle.

## The two putative Lon substrates CCNA_00944 and MotD are part of one single protein that corresponds to FliK

In addition to StaR, our proteomics experiments identified several proteins involved in flagella-based motility and chemotaxis as potential Lon substrates (*Figure 1F*). Particularly promising hits in this group of proteins were CCNA_00944 and MotD (CCNA_00945) that are encoded by partly overlapping open reading frames and showed similar changes in protein abundance and stability in the Lon deficient and Lon overproducing strains in our proteomics experiments (*Figure 4A–B*). While CCNA_00944 is annotated as a flagella hook length determination protein, MotD is annotated as a chemotaxis protein. According to a signature-based annotation of MotD in UniProtKB (entry

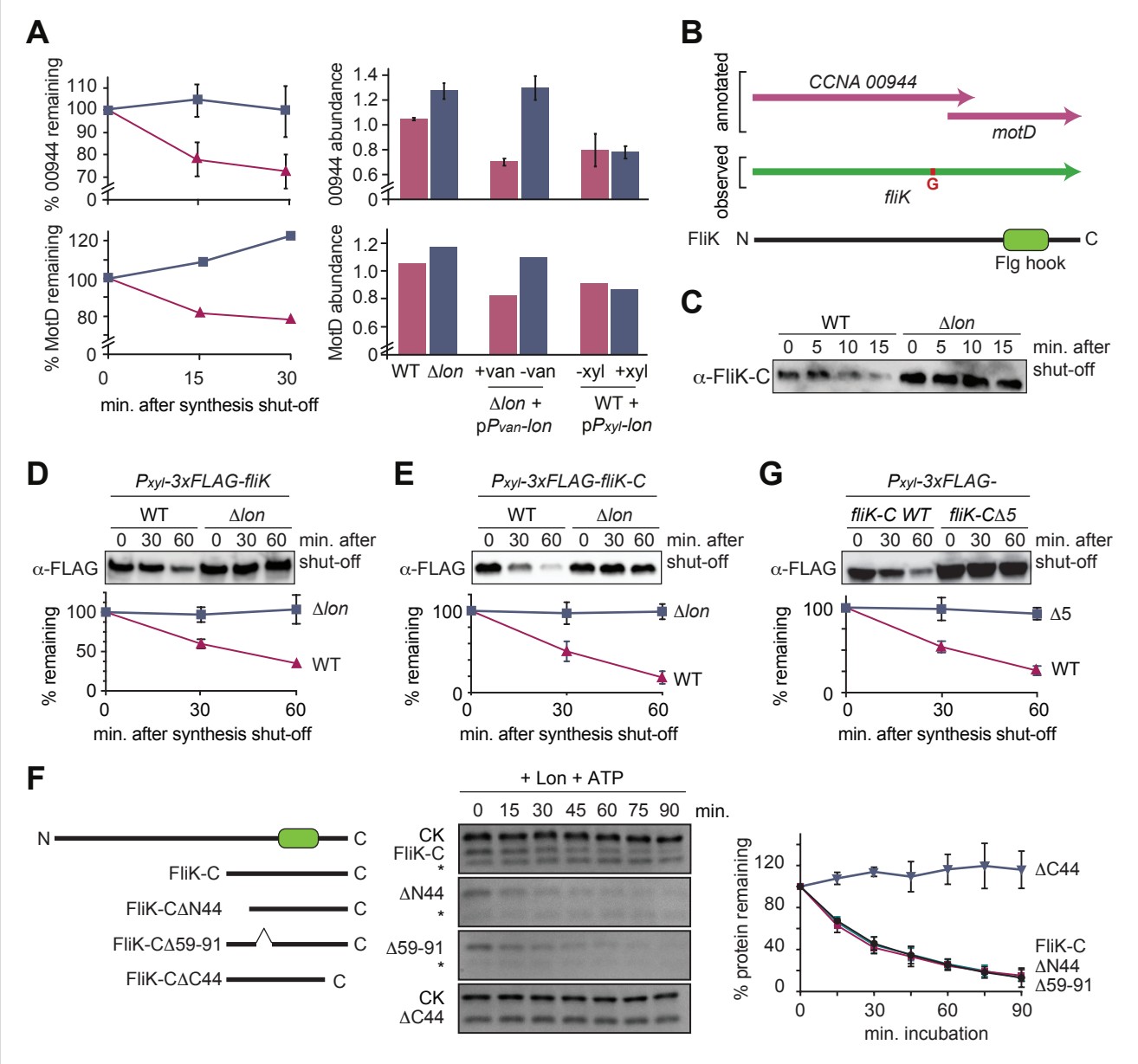

**Figure 4.** The flagella hook length regulator FliK is a Lon substrate with a C-terminal degradation tag. (**A**) Proteomics data obtained for CCNA_00944 and CCNA_00945 (MotD). The graphs on the left show CCNA_00944 and MotD stability in wild-type (WT) and Δ*lon* (LS2382) cells, and graphs on the right show CCNA_00944 and MotD abundance in the different strain backgrounds and conditions as determined by mass spectrometry. Each data point for CCNA_00944 represents the mean protein abundance of the two experimental replicates, error bars show standard deviations. MotD was only detected in one of the replicates. (**B**) Schematic representation of the CCNA_00944 and *motD* (CCNA_00945) genes as annotated in the *Caulobacter crescentus* NA1000 genome, and how the presence of an additional guanosine (highlighted in red) merges the two genes to form one continuous gene that was named *fliK*. The corresponding FliK protein contains a Flg hook domain. (**C**) In vivo degradation assay of full-length FliK in WT and Δ*lon* (KJ546) cells. Samples were taken 0, 5, 10, and 15 min after shutting off protein synthesis. (**D**) In vivo degradation assay of ectopically expressed N-terminally 3xFLAG-tagged full-length FliK in WT and Δ*lon* (KJ546) cells. The graph shows mean values and standard deviation of relative protein levels 0, 30, and 60 min after protein synthesis shut off determined from three independent experiments. (**E**) In vivo degradation assay of the C-terminal part of FliK (FliK-C) in WT and Δ*lon* (KJ546) cells. N-terminally 3xFLAG tagged FliK-C was ectopically expressed and samples were taken at indicated time points after shut off of protein synthesis. Quantifications show mean values obtained from three independent biological replicates and error bars represent standard deviation. (**F**) In vitro degradation assays showing Lon-dependent degradation of FliK-C and truncated FliK-C proteins lacking N-terminal, internal, or C-terminal regions, graphically illustrated on the left side of the panel. Degradation assays were carried out in Lon reaction buffer with 4 μM of one of the FliK-C variants, 0.125 μM of Lon hexamer in the presence of the ATP regeneration system (ATP, creatine phosphate, and creatine kinase [CK]). Asterisks (*) mark a low molecular weight protein, that co-purifies with FliK-C (and some variants) but is unaffected by Lon. The band intensities

*Figure 4 continued on next page*

*Figure 4 continued*
of the FliK-C variants from three independent experiments are represented as means with standard deviations on the right side of the panel. (**G**) In vivo degradation assay of N-terminally 3xFLAG tagged FliK-C (WT) and a variant lacking the C-terminal five amino acids (FliK-CΔ5) after xylose induction in WT cells. Samples were taken at indicated time points after shut off of protein synthesis. Quantifications show mean values of relative protein levels obtained from three biological replicates and error bars represent standard deviation.

The online version of this article includes the following figure supplement(s) for figure 4:

**Source data 1.** Unprocessed Western blot and protein gel images.

**Figure supplement 1.** Western blots of native FliK and 3xFLAG tagged FliK.

**Figure supplement 1—source data 1.** Unprocessed Western blot images.

**Figure supplement 2.** In vitro FliK-C degradation by Lon is ATP-dependent.

**Figure supplement 2—source data 1.** Unprocessed protein gel image.

A0A0H3C6R3), MotD contains a conserved Flg hook domain that is commonly present in the C-terminal portion of FliK proteins that control flagella hook length in many bacteria (*Waters et al., 2007*). When we attempted to clone CCNA_00944, we repeatedly observed one additional guanosine in the cloned gene sequence that was not present in the reference genome sequence of *C. crescentus* NA1000, the strain that we use. This insertion, which is also present in the sequence reads of previously published RNA-sequencing data (*Schrader et al., 2014*), generates a frameshift that merges the CCNA_00944 gene with the downstream located *motD* gene, thus forming one single open reading frame, of which the 3′ portion corresponds to *motD* (*Figure 4B*). This resulting gene corresponds to a single open reading frame (CC_0900) in *C. crescentus* CB15, the isolate from which NA1000 is derived. These observations indicate that CCNA_00944 and *motD* are incorrectly annotated in the reference genome of NA1000 and instead form one continuous open reading frame. Consistently, when we performed Western blot analysis with antiserum raised against the protein portion corresponding to MotD, we detected one single protein band that runs at high molecular weight (*Figure 4—figure supplement 1*), confirming that CCNA_00944 and *motD* form together one single open reading frame. Because the C-terminal portion of this new gene encodes a Flg hook domain that is a characteristic of FliK proteins in other bacteria and because no other gene has so far been annotated as *fliK* in *C. crescentus*, we named the new gene *fliK* (*Figure 4B*). This annotation of *fliK* is also in line with a recent study in *Sinorhizobium meliloti*, which suggested that the *motD* gene from alpha-proteobacteria should be renamed *fliK* (*Eggenhofer et al., 2006*).

## FliK is a Lon substrate that is recognized at its C-terminus

Having established that CCNA_00944 and MotD are part of the same FliK protein, we next investigated its regulation by Lon. Consistent with our proteomics data, we found that FliK abundance and stability were notably increased in Δ*lon* cells, consistent with FliK being a Lon substrate (*Figure 4C*). To investigate the sequence determinants required for Lon to interact with FliK, we expressed an N-terminally FLAG-tagged version of FliK and monitored its stability in vivo. Similar to the non-tagged FliK, 3xFLAG-FliK was efficiently degraded in WT cells but stable in cells lacking Lon (*Figure 4D*). This result led us to hypothesize that the recognition by Lon occurs via the C-terminal domain of FliK. Therefore, we monitored the stability of the C-terminal portion of FliK containing the Flg hook domain, which corresponds to the formerly annotated MotD protein. Like the full-length protein, degradation of this truncated FliK protein (FliK-C), with a FLAG-tag at the N-terminus, depended strongly on Lon (*Figure 4E*). Furthermore, in vitro degradation assays showed that Lon degrades non-tagged FliK-C in an ATP-dependent manner (*Figure 4F*, *Figure 4—figure supplement 2*). Based on these results, we conclude that FliK is a Lon substrate, and that its C-terminal portion containing the Flg hook domain is sufficient for Lon-dependent degradation.

To further pinpoint the regions within FliK-C that are required for Lon-dependent turnover, we analyzed the degradation of a set of additional truncation mutants. According to the MobiDB database (*Piovesan et al., 2021*), the C-terminal portion of FliK lists two unordered regions. We engineered FliK-C variants that lack either of these unordered regions (FliK-CΔN44 and FliK-CΔ59–91) or the C-terminal part (FliK-CΔC44). In vitro degradation assays showed that deletion of the unordered regions did not influence degradation, whereas removal of the 44 C-terminal amino acids completely stabilized the protein (*Figure 4F*). This result, and the fact that some of the degrons recognized by

Lon are located at the very C-terminus of Lon substrates (*Burgos et al., 2020*; *Ishii et al., 2000*; *Puri and Karzai, 2017*; *Zhou et al., 2019*), prompted us to determine the in vivo stability of a FliK-C variant lacking only the C-terminal five amino acids with the sequence LDIRI (3xFLAG-FliK-CΔ5) (*Figure 4G*). This deletion abolished FliK-C degradation (*Figure 4G*), demonstrating that the interaction between Lon and FliK depends on these C-terminal amino acids.

## Lon ensures temporally restricted accumulation of FliK during the cell cycle

Like many other proteins involved in flagella biogenesis in *C. crescentus*, the transcription of the two annotated genes CCNA_00944 and *motD* that together form the *fliK* gene is cell cycle regulated (*Lasker et al., 2016*; *Schrader et al., 2016*), with mRNA levels and ribosome occupancy peaking in late S-phase when a new flagellum at the pole opposite the stalk is being assembled (*Figure 5A*). Consistently, Western blot analysis with synchronized WT *C. crescentus* cultures showed that FliK protein was not detectable in the beginning of the cell cycle, but began to accumulate in late stalked cells before reaching a maximum in abundance in predivisional cells shortly before cell division (*Figure 5B*). This cell cycle-dependent pattern of FliK abundance was completely absent in the Δ*lon* strain, in which FliK was already detectable in swarmer cells and remained at high levels throughout the cell cycle (*Figure 5B*). This result shows that, as in the case of StaR, Lon is absolutely necessary to ensure that the protein is eliminated in the cell cycle phase when its function is no longer needed.

## Precise regulation of FliK abundance is required for proper flagellin expression and flagella function

Next, we wanted to investigate if the Lon-dependent regulation of FliK abundance is required for proper motility in *C. crescentus*. Consistent with a previous study (*Yang et al., 2018*), we observed that Δ*lon* cells show reduced motility in soft agar compared to the WT (*Figure 5C*), which might be caused by reduced flagella function in addition to growth and cell division defects. Additionally, our proteomics data revealed that the levels of α-flagellins FljJ, FljK, and FljL and β-flagellins FljM and FljN, which compose the structural components of flagella, are strongly downregulated in the Δ*lon* mutant (*Figure 5D*). Although a Lon-dependent effect on flagellin levels was not apparent after 4.5 hr of Lon depletion and only to a lesser extent upon *lon* overexpression (*Figure 5D*), these data point to an indirect involvement of Lon in the regulation of flagella biosynthesis. One explanation for the motility defect and the reduced flagellin levels in the Δ*lon* mutant might be the stabilization and higher levels of SciP in this strain (*Figure 1A*; *Gora et al., 2013*), which is known to negatively affect flagellin gene expression through CtrA (*Gora et al., 2010*). Since correct regulation of FliK levels was shown to be critical for proper flagella biosynthesis in other species (*Muramoto et al., 1998*; *Waters et al., 2007*), we thought that the stabilization and thus increased abundance of FliK in the absence of Lon might contribute to the motility defect of Δ*lon* cells as well. To specifically study the consequences of increased FliK abundance, we overexpressed FLAG-tagged FliK, FliK-C, and FliK-CΔ5 from a medium copy vector in otherwise WT cells and assessed soft agar motility and flagellin levels. Overexpression of FliK led indeed to a reproducible reduction in swim diameter to 85% compared to the vector control strain (*Figure 5E*), indicating that elevated levels of FliK impair motility. Interestingly, while this effect was clearly exacerbated in the strain overexpressing FliK-C (*Figure 5E–F*), it was absent in the strain overexpressing FliK-CΔ5, demonstrating that the C-terminus of FliK is critical for the FliK-dependent effect on motility. When analyzing flagellin protein levels in the different overexpression strains, we found that the motility defects of the FliK and FliK-C overexpression strains correlated with a significant downregulation of flagellin levels to 55% and 30%, respectively (*Figure 5G*). Conversely, overexpression of the FliK-CΔ5 affected flagellin levels only mildly (*Figure 5G*).

Taken together, our data indicate that an oversupply of FliK, which can either be caused by overexpression or absence of Lon-dependent degradation, leads to reduced motility and flagellin levels. This suggests that FliK likely contributes to the motility defects of Δ*lon* cells along with SciP and potentially other Lon substrates affecting flagella function that are stabilized and upregulated in the absence of Lon. Furthermore, our data revealed that the C-terminal portion of FliK is critical not only for FliK degradation, but also for its effects on motility and flagellin protein levels.

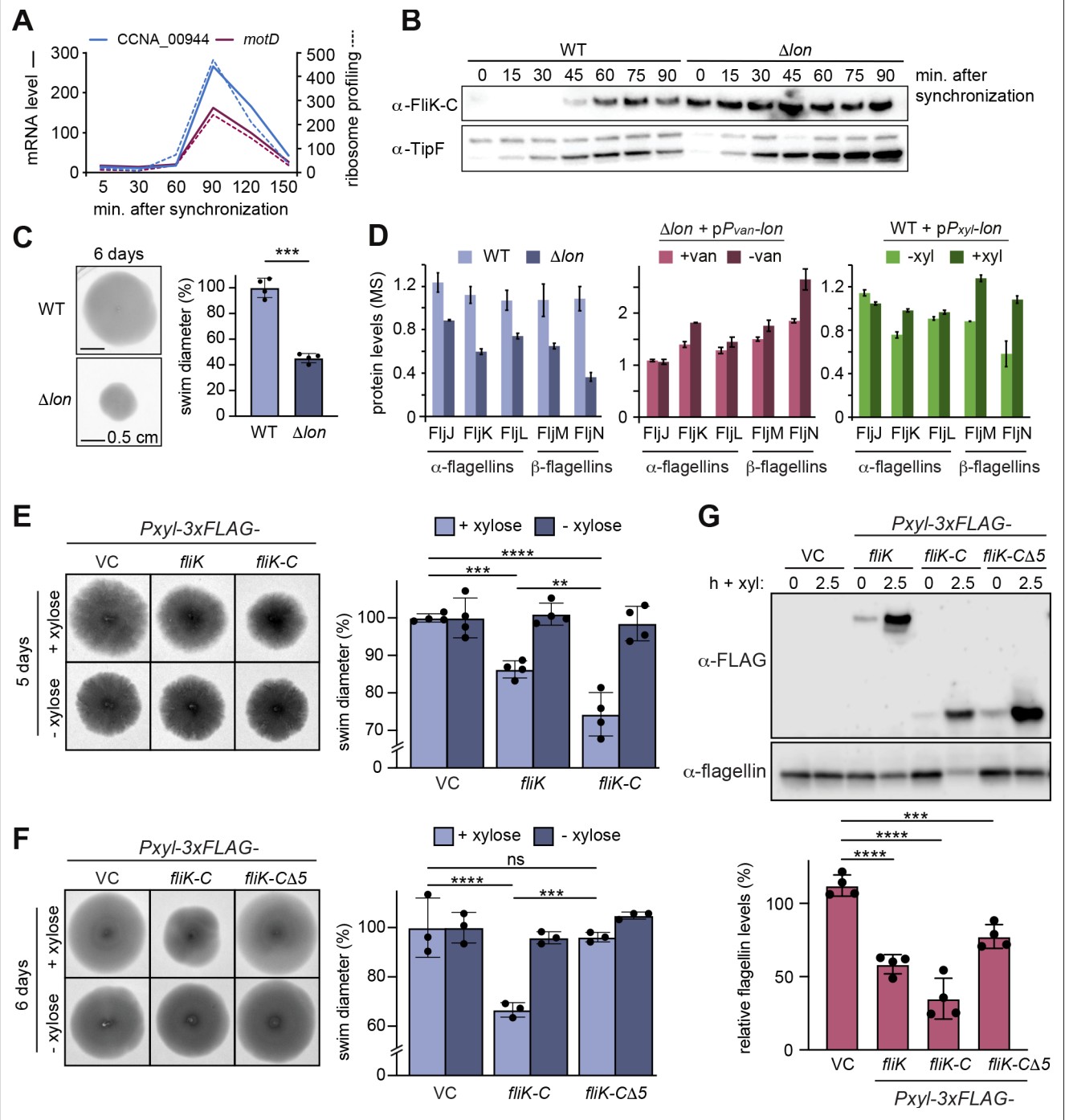

**Figure 5.** Lon-dependent degradation ensures temporal regulation of FliK levels during the cell cycle, which is needed for normal flagellin expression and motility. (**A**) RNA-sequencing and ribosome profiling data for CCNA_00944 and motD (CCNA_00945), as previously published (*Lasker et al., 2016*; *Schrader et al., 2016*). (**B**) Protein levels of FliK and TipF in synchronized wild-type (WT) and Δ*lon* (KJ546) cultures over 90 min following release of swarmer cells into PYE medium. TipF was included as a control. (**C**) Motility assay of *Caulobacter crescentus* WT and the Δ*lon* (KJ546) mutant in PYE soft agar after 6 days. The graph shows the relative swim diameters from four biological replicates, means (mean WT value was set to 100%), and standard deviations are indicated. Statistical significance was determined by paired two-tailed t-test: WT vs. Δ*lon* p=0.0002, ***. (**D**) Flagellin protein levels as determined by mass spectrometry in WT and Δ*lon* (LS2382) mutant cells as well as in the other strain backgrounds and conditions, see *Figure 1*. (**E**) Motility assay in soft agar of strains overexpressing 3xFLAG-tagged FliK and FliK-C by xylose induction (+xylose) in comparison to the vector control (VC) and non-inducing conditions (−xylose). The graph shows the relative swim diameters from four biological replicates, means (mean of VC was set to 100%) and standard deviations are indicated. Statistical significance was determined by ordinary one-way ANOVA (Šidák's multiple comparisons test) for the following comparisons: VC +xyl vs. *Pxyl-3xFLAG-fliK* +xyl p=0.0004, ***; VC +xyl vs. *Pxyl-3xFLAG-fliK-C* +xyl <0.0001, ****; *Pxyl-3xFLAG-fliK* +xyl vs.

*Figure 5 continued on next page*

Figure 5 continued

$P_{xyl}$-3xFLAG-fliK-C +xyl p=0.0016, **. (**F**) Motility assay in soft agar of strains overexpressing 3xFLAG-tagged FliK-C and FliK-CΔ5 by xylose induction (+xylose) in comparison to the VC and non-inducing conditions (−xylose). The graph shows the relative swim diameters from three biological replicates, means (relative to VC) and standard deviations are indicated. Statistical significance was determined by ordinary one-way ANOVA (Šidák's multiple comparisons test) for the following comparisons: VC +xyl vs. $P_{xyl}$-3xFLAG-fliK-C +xyl p<0.0001, ****; VC +xyl vs. $P_{xyl}$-3xFLAG-fliK-CΔ5 +xyl p=0.8184, not significant; $P_{xyl}$-3xFLAG-fliK-C +xyl vs. $P_{xyl}$-3xFLAG-fliK-CΔ5 +xyl p=0.0001, ***. (**G**) Western blot analysis showing total flagellin levels of strains harboring the empty vector (VC) or plasmids for expression of 3xFLAG-tagged FliK, FliK-C, or FliK-CΔ5 before (0) and after induction of expression by xylose for 2.5 hr (lower panel). Induction of the respective FliK variants was determined by Western blot analysis using an anti-FLAG antibody (upper panel). The graph shows the relative flagellin levels compared to the uninduced condition (without xylose) for each strain, as determined by four independent experiments, mean values and standard deviations are indicated. Statistical significance was determined by ordinary one-way ANOVA (Šidák's multiple comparisons test) for the following comparisons: VC vs. $P_{xyl}$-3xFLAG-fliK p<0.0001, ****; VC vs. $P_{xyl}$-3xFLAG-fliK-C p<0.0001,****; VC vs. $P_{xyl}$-3xFLAG-fliK-CΔ5 p=0.0006, ***.

The online version of this article includes the following figure supplement(s) for figure 5:

**Source data 1.** Unprocessed Western blot images.

## Discussion

In all cells, the concentrations of specific proteins must be precisely regulated to maintain cellular functions and to orchestrate complex cellular behaviors in response to external and internal cues. This study uncovered a novel role of the highly conserved protease Lon in coordinating cell differentiation with cell cycle processes in the dimorphic bacterium *C. crescentus*. In this bacterium, each cell cycle phase is coupled to a distinct morphological state (*Curtis and Brun, 2010*). This coupling of cell differentiation with core cell cycle events requires sophisticated mechanisms that coordinate these processes in space and time. Previous work established that in *C. crescentus* large sets of genes are transcriptionally regulated in a cell cycle-dependent manner (*Laub et al., 2000*; *McGrath et al., 2007*; *Schrader et al., 2016*). Using a proteomics approach, we found that several proteins encoded by cell cycle-regulated genes are Lon substrates. The identified Lon substrates include important regulators and structural components required for *C. crescentus* development and cell cycle progression. Our results show that active proteolysis of at least some of these proteins by Lon is required to rapidly clear these proteins following a cell cycle-dependent decrease in their transcription, thus restricting their accumulation to the cell cycle phase when their function is needed (*Figure 6*). The abundance of Lon does not change during the cell cycle (*Wright et al., 1996*) and a recent study suggested that the catalytic activity of Lon is cell cycle independent (*Zhou et al., 2019*). However, it is possible that the degradation of Lon-dependent degradation is affected by the accessibility or functional state of Lon substrates. For example, previous work showed that degradation of both SciP and CcrM is modulated by DNA binding (*Gora et al., 2013*; *Zhou et al., 2019*), and in the case of the AAA+ ATPase DnaA, ATP binding seems to increase protein stability (*Liu et al., 2016*; *Wargachuk and Marczynski, 2015*). Future work will show if similar mechanisms modulate the degradation of the herein identified Lon substrates.

We focused our studies on the stalk regulator StaR and the flagella regulator FliK, which were among the proteins whose abundance and stability were most strongly affected by Lon. Like DnaA, CcrM, and SciP, the transcriptional regulator StaR is a DNA-binding protein. It was initially identified as a positive regulator of stalk biogenesis (*Biondi et al., 2006*) and was later shown to regulate holdfast development by directly inhibiting the expression of the holdfast inhibitor HfiA (*Fiebig et al., 2014*); holdfast is a polysaccharide-rich adhesin that is produced at the nascent stalked cell pole in late swarmer cells and allows *C. crescentus* to attach to surfaces (*Curtis and Brun, 2010*). Stalk biogenesis and surface attachment must be tightly regulated during the cell cycle, in particular, under environmental conditions (*Fiebig et al., 2014*), and our work revealed that Lon-mediated proteolysis contributes to this by regulating the stability of StaR and likely other proteins involved in this process, such as the histidine phosphotransferase ShpA and the polysaccharide biosynthesis protein CCNA_02361 that we identified as putative Lon substrates in our proteomics screen (*Figure 1F*). The gene encoding CCNA_02361 (CC_2278), which was previously shown to contribute to surface attachment (*Sprecher et al., 2017*), shows a similar cell cycle-dependent pattern in mRNA levels and ribosome occupancy as *staR* (*Lasker et al., 2016*; *Schrader et al., 2016*), with the highest mRNA abundance and translation rate in the swarmer state (*Figure 1F*). Thus, Lon may also contribute to temporally regulating the abundance of this protein during the cell cycle.

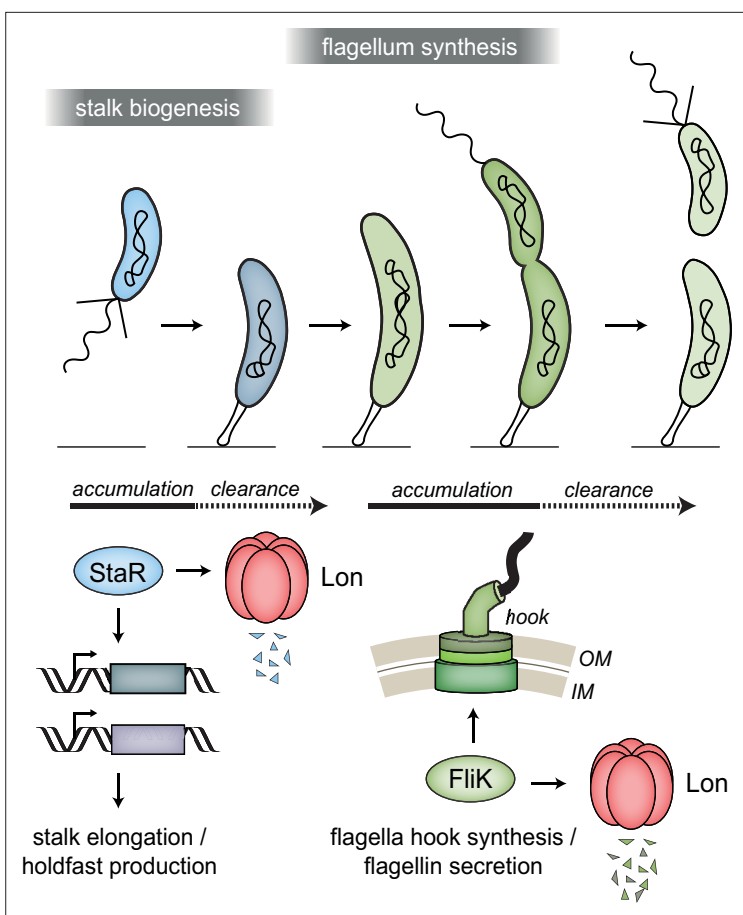

**Figure 6.** Lon ensures temporal regulation of stalk and flagella biogenesis during the *Caulobacter crescentus* cell cycle. Lon specifically degrades StaR, a transcriptional regulator of stalk biogenesis and holdfast production, and FliK, a protein involved in regulating flagella hook synthesis. The expression of *staR* peaks in swarmer cells, while the expression of *fliK* peaks in late stalked and predivisional cells (*Lasker et al., 2016*; *Laub et al., 2000*; *Schrader et al., 2016*). Our study shows that Lon-dependent proteolysis is required to rapidly eliminate these proteins when their expression levels drop, thus outpacing synthesis. The combination of proteolysis and regulated transcription ensures that StaR abundance is temporally restricted to the swarmer-to-stalked cell transition (shown in blue) and FliK abundance to the late stalked and predivisional cell (shown in green) when their functions are needed for stalk biogenesis or flagella synthesis, respectively.

The online version of this article includes the following figure supplement(s) for figure 6:

**Figure supplement 1.** FlgE degradation is partially dependent on Lon.

**Figure supplement 1—source data 1.** Unprocessed Western blot images.

In addition to proteins involved in stalk biogenesis and surface attachment, we identified several proteins required for flagella-mediated motility and chemotaxis as potential Lon substrates (*Figure 1F*), and investigated FliK in detail. FliK proteins are thought to function as molecular rulers that, while being exported themselves, precisely measure flagellar hook length via the N-terminal domain and mediate via the C-terminal domain a switch from export of hook protein to filament protein, once the hook has reached a certain length (*Erhardt et al., 2011*; *Minamino, 2018*; *Minamino et al., 1999*; *Moriya et al., 2006*; *Shibata et al., 2007*). Although previous work made significant progress in understanding the molecular function of this important protein in enterobacteria, proteolytic control mechanisms contributing to FliK regulation have so far not been described. Moreover, in bacteria outside the gamma-proteobacteria FliK remains poorly studied. In *C. crescentus*, a *fliK* gene has previously not been annotated, and our new data revealed that *fliK* corresponds to a gene that was previously annotated as two separate but overlapping genes. We demonstrate that Lon-mediated proteolysis ensures cell cycle-dependent accumulation of FliK during late S-phase, when the cell prepares for cell

division by building a new flagellum and chemotaxis apparatus at the swarmer pole (*Figure 6*). The results that overexpression of FliK results in reduced flagellin levels and impaired motility (*Figure 5E and G*) indicate that this precise regulation of FliK abundance is critical for correct flagella biosynthesis. Based on studies in other bacteria (*Muramoto et al., 1998*), we consider it possible that an excess of FliK protein causes a premature termination of hook synthesis, leading to shorter hooks, which may cause downstream effects, namely reduced flagellin levels and motility. Our observation that overexpression of FliK-C, a truncated version of FliK lacking the N-terminal domain, which is required for export in other bacteria (*Hirano et al., 2005*), exhibits an even stronger effect on flagellin levels and motility may be explained by a dominant-negative effect that the truncated version of FliK has over the WT. In this scenario, overexpressed FliK-C would block the normal function of native FliK, possibly by occupying binding sites on interacting proteins, for example, the export apparatus protein FlhB (*Kinoshita et al., 2017*; *Minamino et al., 2009*). Our result that deletion of the C-terminal five amino acids of *Caulobacter* FliK-C abolishes the negative overexpression effects (*Figure 5F and G*) is consistent with the finding that FliK's export switch-inducing function depends on its C-terminal five amino acids in other bacteria (*Kinoshita et al., 2017*; *Minamino et al., 2006*; *Williams et al., 1996*). Importantly, we also uncovered a critical role of the C-terminal five amino acids of FliK for proteolytic control (*Figure 4G*), thus, our data indicate a tight coupling between FliK function and degradation.

Interestingly, in addition to the flagella hook length regulator FliK, our data indicate that the flagella hook protein FlgE itself is regulated by Lon in *C. crescentus*. FlgE was among the proteins that satisfied two criteria in our proteomics approach, and we verified by Western blot analysis that FlgE degradation partly depends on Lon (*Figure 6—figure supplement 1*). This finding is also consistent with previous work showing that Lon degrades FlgE in *E. coli* (*Arends et al., 2018*), and reinforces the previously made notion that precise regulation of the cellular concentrations of FlgE as well as FliK is an important requirement for the correct temporal order of flagella assembly (*Inoue et al., 2018*).

In several other bacteria, including important human pathogens, Lon has been linked to flagella-mediated motility (*Ching et al., 2019*; *Clemmer and Rather, 2008*; *Fuchs et al., 2001*; *Rogers et al., 2016*). In many of these cases the nature of substrates mediating the observed Lon-dependent effects on motility remains unknown. However, in *Bacillus subtilis*, it was shown that Lon specifically degrades the master regulator of flagellar biosynthesis SwrA by a mechanism requiring the adaptor SmiA (*Mukherjee et al., 2015*). In contrast to SwrA, both StaR and FliK are robustly degraded by Lon in vitro (*Figures 2D and 4F*, *Figure 4—figure supplement 2*). While these data suggest that no adaptors or other accessory proteins are required to mediate the interaction between these proteins and Lon, it is possible that additional factors exist that modulate the rate of Lon-mediated proteolysis of these proteins in response to specific conditions.

In conclusion, our work provides new insights into the cellular roles of Lon and emphasizes the importance of proteolysis in adjusting the amounts of regulatory proteins involved in critical cellular processes, including cell differentiation and cell cycle progression. Importantly, in addition to StaR and FliK our work identified many other proteins as putative Lon substrates and the precise role of Lon in the regulation of these proteins will be worthwhile to investigate in detail in future studies. Furthermore, our work highlights quantitative proteomics using isobaric mass tags as a powerful approach for the identification of novel protease substrates that could be exploited to identify candidate substrates under diverse growth conditions, in different species or of other proteases.

## Materials and methods

**Key resources table**

| Reagent type (species) or resource | Designation | Source or reference | Identifiers | Additional information |
| --- | --- | --- | --- | --- |
| Gene (*Caulobacter crescentus*) | *fliK-C*; *motD*; CCNA_00945 | GeneBank | GeneBank:CCNA_00945 | |
| Gene (*C. crescentus*) | *staR*; CCNA_02334 | GeneBank | GeneBank:CCNA_02334 | |

*Continued on next page*

*Continued*

| Reagent type (species) or resource | Designation | Source or reference | Identifiers | Additional information |
|---|---|---|---|---|
| Strain, strain background (*Escherichia coli*) | DH5$\alpha$ | Other | | Michael Laub, Massachusetts Institute of Technology; Chemical competent cells |
| Strain, strain background (*E. coli*) | BL21-SI/pCodonPlus | Other | | Claes Andréasson, Stockholm University; Electrocompetent cells |
| Strain, strain background (*C. crescentus*) | NA1000 | Other | | Michael Laub, Massachusetts Institute of Technology; Electrocompetent cells |
| Genetic reagent (plasmid) | pBX-MCS-4 (plasmid) | *Thanbichler et al., 2007* | | Martin Thanbichler, MPI Marburg |
| Genetic reagent (plasmid) | pSUMO-YHRC | *Holmberg et al., 2014* | RRID:Addgene_54336 | |
| Antibody | Goat anti-mouse IgG (H+L) Secondary Antibody, HRP | Thermo Fisher Scientific | Cat# 32430; RRID:AB_1185566 | (1:5000) |
| Antibody | Goat anti-Rabbit IgG (H+L) Secondary Antibody, HRP | Thermo Fisher Scientific | Cat# 32460; RRID:AB_1185567 | (1:5000) |
| Antibody | ANTI-FLAG M2 antibody (Mouse monoclonal) | Sigma-Aldrich | Cat# F1804; RRID:AB_262044 | (1:5000) |
| Antibody | Anti-DnaA (Rabbit polyclonal) | *Jonas et al., 2011* | | (1:5000) |
| Antibody | Anti-Lon (Rabbit polyclonal) | Other | | (1:10,000) kindly provided by R.T. Sauer |
| Antibody | Anti-CcrM (Rabbit polyclonal) | *Stephens et al., 1996* | | (1:5000) |
| Antibody | Anti-SciP (Rabbit polyclonal) | *Gora et al., 2010* | | (1:2000) |
| Antibody | Anti-TipF (Rabbit polyclonal) | *Davis et al., 2013* | | (1:5000) kindly provided by P. Viollier |
| Antibody | Anti-StaR (Rabbit polyclonal) | *Fiebig et al., 2014* | | (1:500) |
| Antibody | Anti-FliK-C (Rabbit polyclonal) | This paper | | (1:500) |
| Antibody | Anti-flagellin (Rabbit polyclonal) | *Brun and Shapiro, 1992* | | (1:2000) kindly provided by Y. Brun |
| Peptide, recombinant protein | Ulp1-6xHis | Other | | Source vector and purification protocol kindly provided by Claes Andréasson (Stockholm University) |
| Commercial assay or kit | SuperSignal West Femto Maximum Sensitivity Substrate | Thermo Fisher Scientific | Cat # 34095 | |
| Software, algorithm | Image Lab | Bio-Rad https://www.bio-rad.com/en-ca/product/image-lab-software | RRID:SCR_014210 | Version 6.0 |
| Software, algorithm | GraphPad Prism | https://www.graphpad.com | RRID:SCR_002798 | Version 7.0 |
| Software, algorithm | Fiji (ImageJ) | *Schindelin et al., 2012* https://fiji.sc/ | RRID:SCR_002285 | |
| Other | 4–20% Mini-PROTEAN TGX Stain-Free Protein Gels, 15 well, 15 µl | Bio-Rad | Cat # 4568096 | |
| Other | Trans-Blot Turbo System | Bio-Rad | Cat # 1704150EDU | |
| Other | LI-COR Odyssey Fc Imaging System | LI-COR | | https://www.licor.com/bio/odyssey-fc/ |

## Strains and plasmids

All bacterial strains, plasmids, and primers used in this study are listed in *Supplementary file 1*.

## Plasmid construction

### Expression plasmids for protein purification

Plasmids used for protein expression are based on the pSUMO-YHRC backbone and were constructed as follows: the coding sequences of the *staR* (pMF56-c88) and *fliK*-C (formerly *motD*; pMF61) genes were amplified from *C. crescentus* NA1000 genomic DNA with the primers listed in *Supplementary file 1* (see sheets listing primers and vector fragments for sequences and primer combinations, respectively). The backbone vector pSUMO-YHRC was amplified in two parts disrupting the kanamycin resistance gene in order to reduce background (using primer pairs oMJF34/oMJF36 and oMJF37/oMJF38, *Supplementary file 1*). Following the PCR, the template was digested with DpnI (10 U) and the remaining PCR fragments were subsequently purified by gel extraction. Fragments were then assembled using Gibson assembly (*Gibson et al., 2009*). Vectors containing deletions of an annotated gene (FliK-C truncations: pMF66, pMF67-A, and pMF68-A) were derived from vectors harboring the full-length coding sequence in a similar manner using the primer pairs specified in *Supplementary file 1*.

### Replicating plasmids

pDJO145 (pBX-MCS-4 containing *NdeI-3xFLAG-KpnI*): plasmid pBX-MCS-4 (*Thanbichler et al., 2007*) was amplified using primers oDJO13 and oDJO41. The sequence encoding the triple FLAG tag (*3xFLAG*; GAC TAC AAA GAC CAT GAC GGT GAT TAT AAA GAT CAT GAC ATC GAC TAC AAG GAC GAC GAC GAC AAG) was amplified from a plasmid using primers oDJO42 and oDJO43 adding a KpnI-site followed by a stop codon to the 3′ end of *3xFLAG*. The two amplified fragments were then joined by Gibson assembly (*Gibson et al., 2009*).

pDJO151 (pBX-MCS-4 containing $P_{xyl}$-*staR-3xFLAG*): *staR* was amplified with primers oDJO44 and oDJO45 using chromosomal *C. crescentus* NA1000 DNA as template and cloned into NdeI-cut pDJO145 using Gibson assembly.

pDJO157 (pBX-MCS-4 containing $P_{xyl}$-*3xFLAG-staR*): *staR* was amplified with primers oDJO46 and oDJO47 using chromosomal *C. crescentus* NA1000 DNA as template and cloned into KpnI-cut pDJO145 using Gibson assembly.

pDJO173 (pBX-MCS-4 containing $P_{xyl}$-*flgE-3xFLAG*): *flgE* was amplified with primers oDJO65 and oDJO66 using chromosomal *C. crescentus* NA1000 DNA as template and cloned into NdeI-cut pDJO145 using Gibson assembly.

pDJO200 (pBX-MCS-4 containing $P_{xyl}$-*3xFLAG-fliK-C*): *fliK*-C (*CCNA_00945*) was amplified with primers oDJO87 and oDJO88 using chromosomal *C. crescentus* NA1000 DNA as template and cloned into KpnI-cut pDJO145 using Gibson assembly.

pDJO410 (pBX-MCS-4 containing $P_{xyl}$-3xFLAG-*fliK-CΔ5*): *fliK-CΔ5* was amplified with primers oDJO87 and oDJO179 using *C. crescentus* NA1000 DNA as template and cloned into KpnI-cut pDJO145 using Gibson assembly.

pDJO487 (pBX-MCS-4 containing $P_{xyl}$-*3xFLAG-fliK*): *fliK* was amplified with primers oDJO75 and oDJO88 using chromosomal *C. crescentus* NA1000 DNA as template and cloned into KpnI-cut pDJO145 using Gibson assembly.

## Strain construction

To generate the Δ*staR* Δ*lon* strain (KJ1037), the Δ*staR* deletion was introduced into the Δ*lon* strain (KJ546) by two-step recombination (*Skerker et al., 2005*) after transformation with plasmid pNTPS138-Δ*staR* (pAF491; *Fiebig et al., 2014*). Briefly, transformants were selected on kanamycin plates, single colonies were grown overnight in PYE and plated on PYE containing sucrose. Single sucrose-resistant colonies were subsequently screened for kanamycin sensitivity and the *staR* knockout was confirmed by colony PCR using primers oDJO40 and oDJO38.

*C. crescentus* strains carrying replicating plasmids were created by transforming the plasmids into the respective strain backgrounds by electroporation.

## Standard growth conditions

*C. crescentus* strains were routinely grown at 30°C in PYE medium while shaking at 200 rpm and, if necessary, regularly diluted to assure growth in the exponential phase. If required, the medium was supplemented with xylose (0.3% final), glucose (0.2% final), or vanillate (500 µM final). Antibiotics were used at following concentration (liquid/solid media): gentamycin 0.625/5 µg/ml, chloramphenicol 1/1 µg/ml, and oxytetracycline 1/2 µg/ml. Experiments were generally performed in the absence of antibiotics when using strains in which the resistance cassette was integrated into the chromosome. For phosphate starvation experiments, log-phase cells grown in M2G (minimal medium with 0.2% glucose) were washed and transferred to M5G medium lacking phosphate.

*E. coli* strains for cloning purposes were grown in LB medium at 37°C, supplemented with antibiotics at following concentrations (liquid/solid media): chloramphenicol 20/40 µg/ml, gentamycin 15/20 µg/ml, kanamycin 30/50 µg/ml, and oxytetracyclin 12/12 µg/ml.

## Synchronization of *C. crescentus* cultures

To synchronize *C. crescentus* cultures, cells were pelleted by centrifugation at 8000 rpm for 4 min at 4°C. The supernatant was aspirated, and tubes were kept on ice. Pellets were resuspended in 1 ml of 1× M2 salts on ice. 1 ml of cold Percoll was added and samples were mixed well. The mixture was aliquoted into two Eppendorf tubes and centrifuged at 10,000×*g* for 20 min at 4°C. The top layer of the cells was aspirated, and the swarmer cells were moved into a new tube. Swarmer cells were washed twice with 1 ml of cold 1x M2 salts and finally resuspended in 20 ml prewarmed PYE medium containing antibiotic if required. Samples were taken immediately after resuspending the cell pellet and subsequently at the indicated time points for immunoblot analysis.

## Immunoblot analysis

For whole-cell extract analysis, 1 ml culture samples were collected after the indicated treatments and time points, and cell pellets were obtained by centrifugation. Cell pellets were resuspended in appropriate amounts of 1× SDS sample buffer, to ensure normalization of the samples by units $OD_{600}$ of the cultures. Samples were boiled at 98°C for 10 min and frozen at –20°C until further use. The thawed samples were separated by SDS-PAGE using TGX Stain-free gels (Bio-Rad), and subsequently transferred to nitrocellulose membranes by a semi-dry blotting procedure as per the manufacturer's guidelines. The protein gels and membranes were imaged using a Gel Doc Imager before and after the transfer, respectively, to assess equal loading of total protein as well as the quality of the transfer.

Membranes were blocked in 10% skim milk powder in TBS-Tween (TBST) and protein levels were detected using the following primary antibodies and dilutions in 3% skim milk powder in TBST: anti-CcrM 1:5000 (*Stephens et al., 1996*), anti-DnaA 1:5000 (*Jonas et al., 2011*), anti-Lon 1:10,000 (kindly provided by R.T. Sauer), anti-FLAG M2 antibody 1:5000 (Sigma-Aldrich), anti-SciP 1:2000 (*Gora et al., 2010*), anti-flagellin 1:2000 (kindly provided by Y. Brun) (*Brun and Shapiro, 1992*), anti-FliK-C 1:500, anti-StaR 1:500 (*Fiebig et al., 2014*), and anti-TipF 1:5,000 (kindly provided by P. Viollier) (*Davis et al., 2013*). Secondary antibodies, 1:5000 dilutions of anti-rabbit or anti-mouse HRP-conjugated antibodies (Thermo Fisher Scientific), and SuperSignal Femto West (Thermo Fisher Scientific) were used to detect primary antibody binding. Immunoblots were scanned using a Chemidoc (Bio-Rad) system or an LI-COR Odyssey Fc system. Relative signal intensities were quantified using the Image Lab software package (Bio-Rad) or ImageJ software.

## In vivo degradation assay

To assess protein degradation in vivo, cells were grown under the appropriate conditions (e.g., for 1–2 hr in the presence of xylose to induce expression of FLAG-tagged proteins), and subsequently protein synthesis was shut off by addition of chloramphenicol (100 µg/ml) or tetracycline (10 µg/ml). Samples were taken at the indicated time points and snap frozen in liquid nitrogen before preparation for analysis by Western blot analysis.

## Quantitative proteomics

Sample preparation was performed as previously described (*Schramm et al., 2017*). In brief, two independent cultures for each analyzed condition were harvested by centrifugation. Cell pellets were washed using cold $ddH_2O$ and stored at –80°C. Protein digestion, TMT10 plex isobaric labeling, and

the mass spectrometrical analysis were performed by the Clinical Proteomics Mass Spectrometry Facility, Karolinska Institute/Karolinska University Hospital/Science for Life Laboratory.

To identify putative Lon substrates, first, the protein abundances for each condition within one biological replicate were considered to calculate the following ratios (see *Figure 1—source data 1*): protein levels in Δ*lon* at t=0 min/protein levels in WT at t=0 min (Δ*lon* 0 /WT 0) to identify proteins with a higher steady-state level in the absence of Lon; protein levels after Lon depletion/protein levels before Lon depletion (Lon depletion (no van)/*lon* expression (+van)) to identify differences in protein levels after Lon depletion; protein levels before *lon* overexpression in the presence of glucose/protein levels after 1 hr xylose-induced *lon* overexpression (gluc/+xyl 1 hr *lon* overexpression), to identify proteins that are downregulated by *lon* overexpression; protein levels in WT at t=0 min/protein levels in WT at t=30 min (WT 0 /WT 30), to identify proteins that are degraded in WT cells after protein synthesis shut off. Additionally, we calculated the ratios of protein levels in Δ*lon* cells at t=0 min/ protein levels in Δ*lon* cells at t=30 min after protein synthesis shut off (Δ*lon* 0/Δ*lon* 30). Subsequently, the average of the ratios obtained from the two biological replicates was calculated and used for further analysis. Some proteins, as indicated in *Figure 1—source data 1*, were detected in only one of the replicate data sets. In these cases, only the ratio from the replicate, in which they were detected, was considered.

Next, we selected all proteins with a ratio of (Δ*lon* 0 /WT 0) ≥1.1 for the group 'higher levels in Δ*lon*'. Similarly, we selected all proteins with a ratio of (Lon depletion (no van)/*lon* expression (+van)) ≥1.1 for the group 'higher levels after Lon depletion'. For the group 'lower levels after *lon* overexpression', we selected all proteins with a ratio of (gluc/+xyl 1 hr *lon* overexpression) ≥1.05. For the group 'stabilized in Δ*lon*', we first selected all proteins with a ratio of (WT 0 /WT 30) ≥1. For those, we calculated the ratio of (WT 0/WT 30)/(Δ*lon* 0/Δ*lon* 30) and chose the proteins with a ratio ≥1.05. The low threshold ratios were chosen due to a low dynamic range of the data and to ensure that any promising Lon substrate candidates were not missed. To eliminate false positives that may have passed these individual thresholds, an additional filter was applied that at least three of the threshold ratios needed to be met in order to consider a protein a putative Lon substrate. For this, overlaps between the groups of proteins passing the individual thresholds were determined and graphically displayed using jvenn (*Bardou et al., 2014*; *Figure 1D*). Functional categories were assigned to the group of putative Lon substrates as well as all detected proteins as previously described (*Schramm et al., 2017*).

## Protein purification

Protein purification was adapted from *Holmberg et al., 2014*. In brief, BL21-SI/pCodonPlus cells were transformed using a pSUMO-YHRC derived vector by electroporation. Transformants were selected on LB agar plates lacking NaCl (LBON) supplemented with kanamycin and chloramphenicol and pre-cultures inoculated with 20 colonies before being cultivated at  30°C overnight. Pre-cultures were diluted 1:100 in 1 L 2xYTON and grown to OD$_{600}$ 1.0–1.5. Protein expression was induced with 0.5 mM IPTG and 0.2 M NaCl either for 4.5 hr at either  30°C (StaR and FliK-C) or overnight at  20°C (in case of FliK-C truncations). After centrifugation, cell pellets were stored dry at −80°C.

Pellets were resuspended in HK500MG (40 mM HEPES-KOH pH 7.5, 500 mM KCl, 5 mM MgCl, and  5% Glycerol) supplemented with 1 mM PMSF, 1 mg/ml Lysozyme, and 3 μl Benzonase/10 ml suspension and topped up to 20 ml. Cells were then lysed by 2–4 passes through an EmulsiFlex-C3 high-pressure homogenizer. Lysate was cleared by centrifugation at 32,500×*g* at  4°C for 0.5–1 hr. The protein of interest was bound to 1 g Protino Ni-IDA beads at  4°C/on ice for 30 min. After washing five times with approx. 45 ml HK500MG protein was eluted using HK500MG + 250 mM Imidazole and fractions with protein concentrations ≥1 mg/ml were pooled. For 6xHis-SUMO tag removal, 4 μg/ml Ulp1-6xHis was added and imidazole was removed in parallel by dialysis against HK500MG. Tag depletion (except for StaR) was achieved by binding to 1 g Protino Ni-IDA beads as before and flow through containing purified protein was collected. Protein concentration was then checked via SDS-PAGE (Bio-Rad 4 –20% Mini-PROTEAN TGX Stain-Free Protein Gel) and InstantBlue Protein Stain (Expedeon) and quantified using Bio-Rad ImageLab 6.0.1. Protein samples were aliquoted and stored at −80°C.

## StaR refolding

Because StaR was not soluble in neither of the tested buffer conditions and formed precipitates, the protein was refolded (adapted from *De Bernardez Clark et al., 1999*; *Thomson et al., 2012*). The precipitate was collected (centrifugation at 7197×*g*, 4°C) and solubilized in buffer S (50 mM HEPES pH 8.0, 6 M guanidinium-HCl, 1 mM EDTA, and 10 mM DTT) and the protein concentration was adjusted to 0.2 mg/ml. The protein solution was then diluted with an equal volume dialysis buffer D1 (50 mM HEPES pH 8.0, 2 M guanidinium-HCl, and 2 mM EDTA) and dialyzed against 125 volumes dialysis buffer D1 followed by dialysis against 125 volumes dialysis buffer D2 (50 mM HEPES pH 8.0, 1 M guanidinium-HCl, 2 mM EDTA, 0.4 M Sucrose, 500 mM KCl, and 2 mM DTT). The dialysis buffer was then diluted with one buffer volume of dialysis buffer D3 (50 mM HEPES pH 8.0, 2 mM EDTA, 0.4 M Sucrose, 500 mM KCl, and 2 mM DTT) and dialyzed. This was followed by a final dialysis against 125 volumes of dialysis buffer D3 to remove the remaining guanidinium-HCl. Each dialysis step was carried out at 4°C for approx. 24 hr.

Afterward, insoluble StaR molecules were removed by centrifugation (20,000×*g*, 4°C, 10 min) and cleared refolded StaR was supplemented with additional 2 mM DTT and concentrated using a centrifugal filter with a MWCO of 3 kDa (VWR #516-0227P ) and stored at −80°C . The final concentration was determined by SDS-PAGE using a BSA standard and visualized by InstantBlue Protein Stain (Expedeon).

## FliK-C antibody production

Purified FliK-C was used as antigen to generate rabbit polyclonal antisera (Davids Biotechnologie GmbH).

## In vitro degradation assays

In vitro degradation assays were performed as published previously (*Jonas et al., 2013*). The reaction was carried out in Lon reaction buffer (25 mM Tris pH 8.0, 100 mM KCl, 10 mM MgCl$_2$, and 1 mM DTT) employing 0.75 µM Lon (0.125 µM Lon$_6$), 4 µM substrate (if not stated otherwise), and an ATP regeneration system (4 mM ATP, 15 mM creatine phosphate, and 75 µg/ml creatine kinase). The reaction and the ATP regeneration system were prepared separately pre-warmed to 30°C (approx. 4 min). The reaction was started by adding the ATP regeneration system. Samples were taken at indicated time points and quenched by 1 vol. 2× SDS sample buffer (120 mM Tris-Cl pH 6.8, 4% SDS, 20% glycerol, and 0.02% bromophenol blue) and snap frozen in liquid nitrogen. Samples were boiled at 65°C for 10 min and separated by SDS-Page (Bio-Rad 4–20% Mini-PROTEAN TGX Stain-Free Protein Gel), visualized by InstantBlue Protein Stain (Expedeon) and quantified using Bio-Rad ImageLab 6.0.1. Substrate levels were normalized to Lon and/or creatine kinase levels (in case of '–Lon' samples).

## Microscopy

Cells were fixed by addition of formaldehyde (1% final) to culture samples and stored at 4°C . For visualization, fixed cells were transferred onto 1% agarose pads attached to glass slides, covered with a coverslip, and transferred to the microscope. A T*i* eclipse inverted research microscope (Nikon) with 100× /1.45 numerical aperture (NA) objective (Nikon) was used to collect phase-contrast images. The images were processed with Fiji (ImageJ).

## Stalk length measurements

To quantify the stalk length of cells, microscopy images were analyzed using ImageJ. Briefly, the scale was set to one pixel representing an equivalent of 0.0646 µm. The stalk was then manually selected and length was determined (*Figure 3—figure supplement 1*). For half-automated analysis (*Figure 3E*), the software BacStalk was used (*Hartmann et al., 2020*). The data files containing microscopic images were added to the program, the channel 'phase contrast' was selected, and the scale was set to 1 px≙0.0646 µm. Stalk detection was activated by the setting 'My cells have stalks'. For cell detection, the default settings were used.

## Motility assays

To assess motility, strains were grown in PYE media, supplemented with gentamycin to maintain replicating plasmids when necessary, and cultures were diluted to an OD$_{600}$ of 0.1. Subsequently, 1 µl of

each sample was injected about 2 mm vertically into PYE soft agar plates (0.35%), supplemented with gentamycin or gentamycin and xylose when indicated, using a pipette. The plates were incubated at 30°C and pictures were taken with the setting 'Blots: Colorimetric' using a ChemiDoc (Bio-Rad).

## Acknowledgements

The authors thank Peter Chien and his lab for sharing aliquots of purified Lon and for their help with Lon purifications, members of the Jonas lab for discussions and specifically Roya Akar for technical assistance, Claes Andréasson and his lab for providing the BL21-SI/pCodonPlus strain and the pSUMO-YHRC vector and for their help with the His-SUMO protein purification procedures, Yves Brun and Patrick Viollier for sharing aliquots of antibodies and Sean Crosson for providing plasmids and the Δ*staR* strain. The authors also thank the Clinical proteomics facility at KI/KS for support and advice as well as the Protein Expression and Characterization facility at SciLifeLab for sharing their equipment and providing help with the protein purifications. The study was financially supported by grants from the Swedish Research Council (Dnr. 2016-03300 and 2020-03545), the future leaders grant from the Swedish Foundation for Strategic Research (FFL15-0005), and funding from the Strategic Research Area (SFO) program distributed through Stockholm University.

## Additional information

### Funding

| Funder | Grant reference number | Author |
| --- | --- | --- |
| Vetenskapsrådet | 2016-03300 | Kristina Jonas |
| Vetenskapsrådet | 2020-03545 | Kristina Jonas |
| Swedish Foundation for Strategic Research | FFL15-0005 | Kristina Jonas |
| Stockholm University | SFO funding | Kristina Jonas |

The funders had no role in study design, data collection and interpretation, or the decision to submit the work for publication.

### Author contributions

Deike J Omnus, Matthias J Fink, Conceptualization, Data curation, Formal analysis, Investigation, Methodology, Resources, Validation, Visualization, Writing – original draft, Writing – review and editing; Klaudia Szwedo, Data curation, Formal analysis, Investigation, Writing – review and editing; Kristina Jonas, Conceptualization, Formal analysis, Funding acquisition, Investigation, Project administration, Resources, Supervision, Validation, Visualization, Writing – original draft, Writing – review and editing

### Author ORCIDs

Deike J Omnus ⓘD http://orcid.org/0000-0002-4091-4291
Matthias J Fink ⓘD http://orcid.org/0000-0002-4620-2009
Kristina Jonas ⓘD http://orcid.org/0000-0002-1469-4424

### Decision letter and Author response

Decision letter https://doi.org/10.7554/73875.sa1
Author response https://doi.org/10.7554/73875.sa2

## Additional files

### Supplementary files

- Supplementary file 1. List of bacterial strains, plasmids and primers used in this study.
- Transparent reporting form

## Data availability

All data generated or analysed during this study are included in the manuscript and supporting files.

The following previously published datasets were used:

| Author(s) | Year | Dataset title | Dataset URL | Database and Identifier |
|---|---|---|---|---|
| Schrader JM, Li G, Jonathan WS, Lucy S | 2016 | Ribosome Profiling Reveals Translational Control During the *Caulobacter crescentus* cell cycle | https://www.ncbi.nlm.nih.gov/geo/query/acc.cgi?acc=GSE68200 | NCBI Gene Expression Omnibus, GSE68200 |

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
