## [Decision Letter]

[Editors' note: this paper was reviewed by Review Commons.]

**Acceptance summary:**

Protein turnover is a basic feature of life. The Lon protease, a component of the protein degradation machinery, is conserved in all three domains of life. Yet its protein substrates remain poorly characterised. Here the authors present a proteomic-based approach to the discovery of such substrates in the model bacterial organism *Caulobacter crescentus*. Two substrates have been characterised in considerable detail, highlighting Lon's role in bacterial physiology.

---

## [Author Response]

Reviewer #1 (Evidence, reproducibility and clarity (Required)):The Lon protease is highly conserved across 3 domains of life, yet in most organisms very few substrates are known. In this study, the authors build on observations of Lon-dependent protein stability modifications to set up a proteomics-based screen for new Lon substrates in the model bacterium *Caulobacter crescentus* (in which only 3 substrates had been identified so far). They focus the rest of their work on validating and characterising further 2 Lon substrates, StaR (a transcriptional regulator important for stalk biogenesis) and FliK (a regulator of flagellar hook assembly). They were able to confirm that these two proteins are Lon substrates, both in vivo and in vitro (the latter showing that the effect of Lon on the levels of these proteins is not an indirect effect). They could also identify portions of these substrates that are required for Lon-mediated degradation. In the case of FliK, they found that the last 5 residues at the C-terminus of the protein are needed for Lon-dependent degradation and are also important for the effect of FliK on flagellins amounts in the cell. Along the way, they corrected a misannotation of the ORF encoding that protein. This suggest a hypothesis for how the Lon-dependent degradation of FliK might lead to flagellin misregulation, therefore affecting bacterial motility. In the case of StaR, the authors show that the stalk length defects previously observed in the ∆lon mutants are attributed to elevated levels of StaR during the cell cycle. Overall, this is a carefully conducted study and a clear manuscript. The text and figures are easy to understand and the story flows. The abstract and introduction are engaging for a broad audience.Major comments:The authors suggest the possibility that cell-cycle-dependent Lon-mediated degradation of Lon substrates might depend on their functional state, such as their DNA-bound state. This is a very attractive hypothesis considering that the 3 previously identified Lon substrates in Caulobacter are DNA-binding proteins, as well as StaR that they identified here as another Lon substrate. In the discussion, the authors also mention the possibility of accessory proteins to regulate Lon activity. However, I miss the information about Lon protein levels during a (synchronized) cell cycle (either a ref to published data or a new experiment), as this might be constitute another factor for cell cycle regulation and contribute to understanding how Lon can impact protein levels of its substrates in a cell-cycle dependent manner.

According to a previous study by Wright et al. 1996, Lon levels do not change in a cell cycle-dependent manner. As the reviewer suggested, we have added this information to the discussion in l. 329-335.

Related to this, the authors refer to a paper showing that Lon catalytic activity does not seem to change during the cell cycle, in the discussion part, but this could be developed a bit and presented earlier in the introduction.

In this mentioned study by Zhou et al. 2019, it was shown that Lon-dependent degradation of a constitutively expressed model substrate was not affected by the cell cycle phase, which led to the conclusion that Lon activity is cell cycle independent. We have modified the section about this previous work and also added a sentence to mention that future work will be dedicated to mechanisms regulating the degradation of the newly identified substrates. Given that our study was primarily focused on the identification of new Lon substrates and their contributions to Lon-dependent phenotypes, we think this information is better placed in the discussion than in the introduction.

Finally, in their legend of the model figure (Figure 6), the authors suggest that the Lon protease acts on its substrates when their expression level drops. Do the authors imply some sort of stoichiometry at play between Lon and its substrates that would determine when Lon will have an overall impact on their protein levels, or do they consider a more complex mechanism? It would be interesting to discuss this a bit more (I understand that going in this direction experimentally would be another story, and not required to support the data shown here).

Based on the reviewer’s comment, we realized that the figure legend might have been misleading. We do not think that Lon only acts on these substrates when their expression levels drop, but rather that Lon-mediated proteolysis in combination with regulated transcription ensures rapid clearance of the proteins. We have modified the text of this figure legend to enhance clarify (l. 1092).

Minor comments:- Figure 1E: Optional: It might be informative to add the level of enrichment of these categories compared to the whole proteome content.

We thank the reviewer for this suggestion. In the new version of our manuscript, we included, for comparison, a graph to Figure 1E that shows how all detected proteins sort to the different functional categories. To fit both graphs into the main figure, we replaced the pie chart with stacked bar graphs. We have also modified the text (l. 125-130) to emphasize the comparison of protein factions between whole proteome content and the pool of Lon substrates.

– Line 193: the ∆staR∆lon mutant phenocopies the ∆staR single mutant regarding stalk length. I would suggest to add the result of the ANOVA for this comparison on Figure 3 (as done in the related Figure 3—figure supplement 1 showing non-significant difference between these mutants).

As suggested by the reviewer, we have added the results of the statistical tests for the remaining comparisons to the figure and figure legend.

– Figure 4F: I could not find the explanation for the asterisk next to the band below FliK-C. Please add it in the legend.

Thanks for pointing this out. We have added the explanation for the asterisk to the figure legend.

– Figure 4F: did the authors test Lon-dependent degradation in vitro for the FliK-C∆5 mutant as well? They show degradation in vivo only (Figure 4G).

We did not test the degradation of the FliK-C∆5 mutant in vitro. However, based on our in vivo stability data of this mutant and our in vitro data of the FliK-CΔC44 mutant (which shows complete stabilization), we expect this mutant to be stable.

– Line 286: at this point, I would consider that the authors have shown misregulation of flagellin protein levels, not flagellin expression (transcriptional).

We agree with the reviewer that it is better to refer to “flagellin protein levels” rather than “flagellin expression” and changed the wording accordingly.

– Figure 5D: results from the three different lon mutant strains seem contradicting, as Lon deletion and Lon depletion have opposite effects on flagellins protein levels, and Lon overproduction gives results similar to Lon depletion. Could it be that -van and +van labels were inadvertently swapped?If yes, please correct.If not:While the authors appropriately use the term "misregulated" to describe the flagellins in the various lon mutants, the absence of comment on this discrepancy at that place in the text might bring confusion about their interpretation of these results. Adding a clarification here might also help introducing more clearly the experiments that come next.

The Lon-dependent effect on flagellin levels is indeed not apparent in the Lon depletion strain and only to a lesser extent in the Lon overexpression strain. Although we cannot fully explain this difference between the ∆*lon* strain and the conditional Lon strains at this point (see also our response to reviewer 4), we still think that the flagellin data, along with the motility data, support an indirect involvement of Lon in flagella regulation. As the reviewer suggested, we have described the flagellin data more explicitly in the text. Based on the comments by reviewers 3 and 4, we have also restructured this section to enhance clarity.

Related to this, at lines 303-304 (Figure 5F-G), the authors conclude that the reduced flagellin levels and motility defects of ∆lon can at least in part be attributed to the stabilisation of FliK in these cells. Here as well, it might be worth giving a few words on why Lon depletion data (Figure 5D) do not fit with this idea.

In our new version we have improved the section describing the link between FliK, Lon and motility (see comments below). With these changes we hope that we satisfy this point by reviewer 1.

– Figure 5E: please clarify whether a specific a-flagellin was monitored here or all of them.

The antiserum that we used detects all flagellins. We have modified the text and added a reference to indicate the source of this antibody.

– Figure 5E: how do the authors explain that FliK-C overproduction has a stronger effect on a-flagellin levels than the full-length FliK (as both are supposed to be able to interact with FlhB – cfr their hypothesis for the importance of the last 5 aa at lines 292-294).

It is indeed striking that overexpression of FliK-C results in a stronger phenotype than overexpression of the full length FliK protein. We consider it possible that the truncated version of FliK has a dominant negative effect over the wild type. In this scenario, overexpressed FliK-C would block the normal function of native FliK, possibly by occupying binding sites on interacting proteins, such as the export apparatus protein FlhB (Kinoshita et al. 2017). We have added a paragraph to the discussion in which we discuss this possibility (l. 371-383)

– Methods: could the authors explain how they chose the 1.1 or 1.05 ratios (lines 526-534), which may seem quite low?

The main reason for these low thresholds was the low dynamic range of our proteomics data that is due to technical aspects of the TMT-labeling approach as well as our experimental conditions. Even the well-studied substrates, DnaA, SciP and CcrM, that we used as positive controls, showed only small fold changes (1.1-1.7 fold) in the different strain comparisons. Because we did not want to miss promising Lon substrates when setting our filters for the individual conditions, we decided to choose the low thresholds of 1.1 in the *lon* deletion and depletion experiments and 1.05 in the stability assay and overexpression experiment (in these latter experiments the dynamic range was even smaller due to the early time points that we used).

Importantly, to filter out the false positives that may have passed these low thresholds in the individual conditions, we applied the additional requirement that at least three of the thresholds needed to be met in order to consider a protein a putative Lon substrate. With this multi-layered filtering approach, we were able to identify 146 proteins that included the known substrates DnaA, SciP and CcrM as well as the validated new substrates StaR and MotD/CCNA_00944. Based on these results we are confident that our group of putative Lon substrates contains a high rate of true Lon substrates. Nevertheless, we think it is important to perform follow-up experiments with the candidates generated in our study. For this reason, we were careful to call unvalidated hits “putative Lon substrates”. Furthermore, we have presented our proteomics data in a transparent and comprehensive way in dataset 1 that will allow other researchers to carefully evaluate Lon-dependent effects on other proteins of interest.

Based on this comment and the comments by reviewer 2 we have added some more explanation and details to the description of our proteomics data analysis in the Methods section.

Reviewer #1 (Significance (Required)):The Lon protease is highly conserved across 3 domains of life, yet in most organisms very few substrates are known. This work advances the fields of protein homeostasis / bacterial cell biology by identifying new substrates of Lon, using the bacterium *Caulobacter crescentus* (a model for bacterial cell cycle control, development and differentiation) using an interesting proteomic approach. Two of these substrates are characterized further, StaR and FliK. This work certainly highlights yet another layer of sophistication in the complex cell cycle control mechanisms in *Caulobacter crescentus*. While here the authors present Lon as an important factor for regulating/coordinating cell cycle events, how exactly Lon-mediated degradation can be cell-cycle regulated in *Caulobacter crescentus* (at least for the substrates characterized in this study) is an important question that future studies will need to address.In my opinion, the audience will include researchers involved in protein homeostasis (whether working directly on Lon or on other protein degradation/stability systems), bacteriologists (including the broad field of cell cycle regulation), and of course more specifically all researchers working on *Caulobacter crescentus*.My field of expertise is bacterial cell biology in general, including bacterial cell cycle control, intracellular processes, live microscopy imaging and protein localisation in *Caulobacter crescentus* and other species. I do not have sufficient expertise to evaluate the methodological details of the identification and quantification of substrates by mass-spectrometry.Reviewer #2 (Evidence, reproducibility and clarity (Required)):Omnus et al. utilized a quantitative proteomics approach to globally identify novel substrates of Lon protease in the bacterium *Caulobacter crescentus*. Among candidate protease substrates revealed by large proteomic datasets, the authors focused on the developmental regulator StaR and the flagella hook length regulator FliK as direct Lon substrates. With a number of elegant follow-up experiments, their findings established a critical role of Lon in coordinating developmental processes with cell cycle progression. Overall, this work is carefully designed and well performed. The reviewer would recommend its publication upon addressing a few minor issues, which are mostly associated with the quantitative analysis of proteomics data.1. Typically three biological replicates would be needed for proteomic experiments, thus permitting statistical analyses of differentially regulated proteins.

We are aware that two biological replicates are not sufficient to do sophisticated statistical analyses. Therefore, we did not perform a more in-depth analysis that would provide statistical significances on the observed changes. Instead, we treated the proteomics results as a list of potential substrate candidates that provide the basis for rigorous further investigation and validation.

2. Line 524, some proteins are only detected in one replicate and should not be used for quantitative analyses. Given that only two replicates were considered, it is important to stay on the safe side and control the rate of false positives.

We agree that it is important to exclude false positives, and our approach to consider only proteins as putative Lon substrates that satisfied a combination of different criteria helped us certainly to do so. Our decision to include also proteins that were detected in only one of the replicates was taken not to miss any promising Lon substrate candidates. Note that for example CCNA_00945, i.e., MotD, which turned out to correspond to FliK-C, one of the validated new Lon substrates, was detected only in one of the replicates. We also want to stress that one replicate in the proteomics analysis included in fact data points for three independent strain comparisons (1. WT/∆*lon* stability and steady state levels, 2. Lon depletion, depleted vs. non-depleted, 3. Lon overexpression, induced vs. uninduced). Comparing the results from these different experiments against each other gave us good indications regarding the validity of our data.

3. Line 526, the cutoff of the protein ratio is 1.1 (or 1.05 in some cases), which is way too low to tease out the altered proteins……The author should apply the threshold of 1.5 at least.

Please see our response to the last comment by Reviewer #1, in which we have explained why we use these low threshold ratios in our data analysis. Applying a threshold ratio of 1.5 would exclude essentially all proteins from our list of putative Lon substrates including the known substrates DnaA, SciP, and CcrM and the validated substrates StaR, MotD, CCNA_00944. Hence, a threshold of 1.5 was not appropriate for our dataset.

4. When reporting protein fold changes (in the supplemental dataset), two decimals are sufficient. The authors used eight decimals, which do not make any sense.

We agree and have changed the column type to “Number” and set the amount of shown decimals to two decimal places, in order to preserve the full numbers while increasing readability.

Reviewer #2 (Significance (Required)):The findings should add significantly to our growing knowledge about the Lon protease, as well as the regulation of critical biological processes by intracellular proteolysis.Reviewer #3 (Evidence, reproducibility and clarity (Required)):In this work, the authors use a combination of proteomics to identify novel Lon substrates in the bacterium *Caulobacter crescentus*. They focus their search on substrates involved in cell differentiation and identified two new Lon substrates, StaR and FliK, which they show are degraded in a Lon-dependent manner in vivo and are directly proteolyzed by Lon in vitro as well. The authors go on to show that these substrates seem to be recognized by Lon from their C-termini and that Lon-mediated degradation of these substrates allows them to oscillate during the cell cycle. Finally, the authors link misregulation of these substrates in cells lacking Lon with defects in stalk biogenesis and motility. The authors also take the extra effort to clarify the annotation of the fliK gene, which is appreciated. Overall, this is a very strong manuscript, with some concerns as raised below.1. The authors should explicitly state on line 188-191 that overexpression of StaR was shown to increase stalk length (Biondi, et al. 2006) as this is a key premise for their model that stabilization of StaR explains in the increased stalks of ∆lon. As written, it is unclear that StaR has been previously shown to be a positive inducer of stalk length.

We completely agree with this point and have modified the text to mention more explicitly that *staR* overexpression causes stalk elongation and that our experiment is based on this finding (l. 191).

On a related note, the implication of the model is that o/exp of StaR in a ∆lon would not increase stalk length. Is that true?

According to our model, Lon regulates stalk length via StaR, which makes it unlikely that *lon* deletion would diminish the effect of StaR overexpression. Rather, we consider it possible that *staR* overexpression in ∆*lon* cells would lead to even stronger StaR accumulation, which in turn might be reflected in even longer stalks. Because we think that this experiment is not really needed to support the main conclusions of our paper, we have not included it in our study.

2. The authors show convincing data that FliK is a Lon substrate. However, the data tying FliK stabilization to motility defects in a Δlon strain is not as compelling, especially as motility in this media is a combination of cell growth, division, and actual motility. Ideally, the authors could show that flagellar function itself is affected by using chemotaxis assays or something similar. If this cannot be experimentally addressed, then there should be a clear discussion as to the reservations with interpreting these swimming assays in light of the combination effects that change growth.

We agree with the reviewer that the swim diameters in low percentage agar are likely to be affected by changes in growth rate and cell division rate. For this reason, we decided to also include the quantification of flagellin levels in the different strain backgrounds, as we thought that the observed misregulation of flagellin levels further indicates an indirect involvement of Lon in flagella regulation. However, we agree that the limitations of the motility assays should be mentioned in our manuscript and have added a sentence in l. 283 to indicate that growth and division defects may contribute to reduced motility.

3. Why did the authors not determine if deletion of fliK could reverse ∆lon motility defects similar to that shown for StaR?

Studies in other bacteria have shown that FliK is an essential regulator of motility that is critical for proper hook synthesis and filament protein secretion. The deletion of *fliK* or absence of functional FliK leads to elongated hooks without attached filament structure, i.e., to an immotile polyhook phenotype (e.g.*,* Muramoto et al. 1998, Eggenhofer et al. 2006). We consider it likely that deletion of *fliK* in *Caulobacter* will have a similar effect. Therefore, we do not expect to see a reversion of the motility defects when knocking out *flik* in a ∆*lon* strain.

To studying the link between Lon and FliK, we decided to analyze the consequences of FliK overexpression in otherwise wild type cells, thus mimicking the stabilization and higher abundance of FliK that occurs in ∆*lon* cells, while eliminating the effect of Lon on other substrates. Our data indicate that increased FliK abundance has a negative effect on motility and highlights the importance of precisely regulating FliK levels.

The case of StaR is different: though it is a positive regulator too, it is not essential for stalk biogenesis. The stalks of ∆*staR* cells are shorter than WT stalks but they are not completely absent.

4. We found it surprising that there was not more description of how stabilization of SciP (a known Lon substrate) could be contributing to the flagella phenotypes, given that it seems to directly repress expression of many flagella genes. It seems likely that overabundance of SciP underlies why FliK stabilization might only partially explain the ∆lon motility defects.

We agree with the reviewer that stabilization of SciP in *∆lon* cells likely contributes to the motility phenotype. As the reviewer suggested, we discuss this idea more explicitly in the new version of our manuscript (l. 289-292).

5. In Figure 5C, the quantification of the ∆lon swim diameter should also be shown, so that it can be compared with those diameters shown in F and G, especially as the ∆lon swim diameter seems much smaller.

We have added the quantification including statistical analysis of the relative swim diameters to Figure 5C and modified the figure legend accordingly.

There also should be some statistical treatment of the measurements in F and G to show they are different.

We have modified the graphs of Figure 5E and F (formerly Figure F and G) to include the results of the statistical analysis (see also figure legends). For consistency, we have also added statistical analysis of the data displayed in Figure 5G (formerly Figure 5E).

Finally, why were the panels in F and G taken at different days of growth?

The experiments shown in panels 5E and F (formerly Figure 5F and G) were not conducted side by side, hence we happened to photograph the plates after slightly different incubation times (5 or 6 days of incubation at 30°C). Both experiments include a vector control used for normalization, and the relevant experimental samples to allow comparisons and thus can stand for themselves.

6. We found the connection to α and β flagellin levels in Figure 5 confusing. The authors reason that FliK overabundance reduces flagellin levels but in Figure 5D, the authors show that while flagellin levels decrease in a Δlon strain (which is likely also due to SciP stabilization as described above) this is not the case for the Lon depletion strain which would presumably have elevated FliK levels as well. At a minimum, this should be pointed out and discussed in the text.

As explained in our responses to reviewer 1, we have improved and restructured the text describing the data shown in Figure 5D to better explain the link between FliK, flagellins, Lon and motility.

7. It is not clear from the methods how loading of the westerns was controlled. This is particularly important given the reliance on quantification for the in vivo degradation measurements.

We verified equal loading and the quality of transfer by visualizing total protein using TGX Stain-free protein gels (Bio-Rad) prior to and after blotting. We have added this information to the description of immunoblotting in the Materials and methods part of the manuscript.

Reviewer #3 (Significance (Required)):This work reveals the substrate profile of an important protease. It demonstrates both the caveats and advantages of a multi-proteomic strategy. One major addition is the identification of the StaR degradation, which explains some phenotypes of the ∆lon mutant. Overall, this work will be highly valuable to those interested in post translational control in bacteria.Reviewer #4 (Evidence, reproducibility and clarity (Required)):Using a proteomics approach the authors find that the Lon protease regulates the stability of hundred proteins including the StaR stalk transcriptional regulator and the FliK flagellar protein. They show that the degradation of the two proteins by Lon is direct using in vitro assays.The manuscript is clearly written and the data is solid and the experiments are well executed. The data is displayed clearly, sometimes in more than one (redundant) panel. There might be problem with Fig5D, middle panel (see below).The major shortcoming is that this work lacks conceptual novelty: it does not advance the field substantially and it does not establish (convincing) causality between Lon, motility, FliK and the flagellins.

We want to emphasize that our study includes a comprehensive search for Lon substrates in one of the primary bacterial model organisms*.* This search yielded a list of more than 100 Lon substrate candidates, of which we have characterized StaR and FliK as Lon substrates in detail. We were able to demonstrate that Lon-dependent degradation is critical for the cell cycle-dependent regulation of these proteins and for proper control of stalk length and motility. We think our findings will appeal to a wide readership, including scientists studying proteolysis, cell cycle regulation as well as bacterial motility, and reviewers 1-3 seem to agree with this.

We admit that the part regarding FliK, Lon, flagellins and motility might have been somewhat unclear in the first version of our manuscript, which may have caused confusion. To improve our paper, we have made some larger text changes to this section and have also added a new paragraph to the discussion, in which we explain the FliK and FliK-C overexpression effects. Finally, we believe that the discovery and initial characterization of a FliK protein potentially functioning as a hook length regulator in *Caulobacter* is of high relevance. In particular our finding that FliK is proteolytically regulated will likely be appreciated by all groups working on bacterial motility/flagella assembly and function.

Reviewer #4 (Significance (Required)):StaR has been studied before, its role in stalk length control and its transcriptional profile has been determined. Here the contribution is that StaR is regulated at the level of stability by Lon and that the StaR accumulation profile during the cell cycle (in G1 phase) matches that of its transcriptional profile (reported previously), simply because StaR is unstable protein and that this instability is mediated by Lon. If StaR levels are elevated by preventing its degradation, then cells have longer stalks. However, despite this causality,there is no new insight on StaR function in this study. StaR overexpression defects have been described before, and this work only shows that such overexpression can also arise by interfering with degradation.

The main goal of our study was to identify novel Lon substrates and to study the role of Lon in regulating these proteins in order to gain molecular insights into the general mechanisms by which Lon impacts critical cellular processes such as cell differentiation and development. The intention was not to in detail characterize the specific functions of individual Lon substrates, such as StaR, as this would have exceeded the scope of this paper.

The part of FliK is enigmatic. The two first panels of figure 5D appear inconsistent regarding the abundance of flagellins (in panel 2: why are the flagellin levels higher in Δ-lon cells harboring Pvan-lon in the absence of inducer).

We agree that the discrepancy between the *∆lon* and Lon depletion data is unexpected and at this point we can only speculate about the underlying reasons. Given that Lon affects flagellin protein levels most likely indirectly, it is possible that the time point of 4.5 hours that we chose in our depletion experiment was too short to detect an effect on flagellin levels. Despite this apparent inconsistency, we still think that the altered flagellin levels in the *∆lon* and *lon* overexpression strains support an indirect role of Lon in regulating flagella-based motility. In the new version of our manuscript, we describe the flagellin data in more detail.

There is no null phenotype of FliK reported and the observed effect on flagellins could simply arise from a block of secretion caused by the absence of Lon which feeds back on flagellin expression. These effects are well known.

We completely agree that the changes in flagellin levels are likely caused by indirect effects that arise from problems in secretion. In the new version of our manuscript, we discuss this possibility in the discussion (l. 371-373).

It is also not clear that a 50% reduction in flagellins will block motility to the extent seen here or if this is due to other effects of Lon. Causality is missing. Can flagellin overexpression restore motility and what are the effects of deleting fliK also in the Lon mutant background?

The stabilization of FliK in *∆lon* cells is unlikely to be the only cause of the motility defect of these cells. As explained in our responses to reviewer 3, we think that stabilization of SciP and potentially other Lon substrates with functions in flagella assembly / regulation likely contribute to this phenotype. In our new version of this manuscript, we mention this possibility more explicitly in l. 289-292 and l- 309-310. We have also restructured the text dealing with this aspect to improve clarity.

Because we think that the altered flagellin levels in the FliK overexpression strains are an indirect consequence of defects during the flagella assembly process (i.e.*,* premature termination of hook length and secretion problems), we consider it unlikely that flagellin overexpression would restore motility in a ∆*lon* mutant. Although our FliK overexpression data suggest that FliK contributes to the motility phenotype of ∆*lon* cells, we agree that our data do not completely prove causality and we have carefully chosen our wording to not imply this.

Regarding the effects of deleting *fliK* in the Lon mutant background, please see answer to reviewer 3, question 3.

As it stands here, the work only shows that FliK (whose function is unknown in Caulobacter) is degraded by Lon. FliK is cell cycle regulated, but this may again arise from temporal control at the level of synthesis of an unstable protein that is constitutively degraded by Lon.

We disagree that our work “only shows that FliK is degraded by Lon”. Our work is the first study describing a FliK protein in *Caulobacter* and provides an important first characterization of this protein: we show that it accumulates in a cell cycle phase-specific manner and that Lon is required for this. Furthermore, our detailed mutational analysis in combination with the study of FliK-dependent phenotypes uncovered a critical dual role of the C-terminus of FliK in Lon-dependent degradation and FliK function. We agree that our study raised many new questions regarding this interesting protein. However, answering all these questions would certainly go beyond the scope of this manuscript.

Why FliK needs to be degraded seeing that it is likely a protein that is exported by the flagellar system and whether this export and its function is important remain unclear.

These are indeed interesting and very relevant points. It is noteworthy that the cytoplasmic abundance of many other secreted flagella components is precisely regulated by transcriptional, translational and in some cases by proteolytic control mechanisms. It has been proposed by others (e.g., Bonifield et al. 2000, J Bact) that precise regulation of secreted flagella components ensures correct order of secretion and may help to avoid competition between secreted flagella subunits during the assembly process. It will be interesting to address these hypotheses in future studies.

It is also unclear how FliK would regulate flagellins and whether this can impact motility.

As explained above, we think that the effect of FliK on flagellin levels is a consequence of improper secretion that feedbacks to flagellin expression levels. Our data showing that FliK and FliK-C overexpression results in reduced swimming, demonstrate that FliK impacts motility. Based on FliK studies in *Salmonella* (e.g., Muramoto et al. 1998, JMB), we think that the main reason for this result is the premature termination of hook synthesis in the presence of excess FliK, leading to shorter hooks, which may cause downstream effects, i.e., reduced flagellin levels and motility. We have added a new paragraph to our discussion (l. 371-385) in which we discuss this possibility.

In sum, Lon constantly degrades proteins, also those that are cell cycle regulated at the level of transcription/synthesis. Lon may simply fulfill a passive role in promoting the instability of many cytoplasmic proteins, including those that are cell cycle regulated. Thus, it may serve as a factor that renders a selection of cellular proteins unstable, perhaps to ensure that their presence in the cell cycle is determined when they are transcribed, but I think this global role is not sufficiently developed here.

This is the essence of the model we describe in this manuscript and that is illustrated in Figure 6, however we want to emphasize that, according to our data, Lon does not “simply fulfill a passive role” but has a crucial active role in establishing the fluctuation in levels of important proteins encoded by cell cycle regulated genes. We show that in the absence of Lon-dependent proteolysis, the cell cycle-dependent transcriptional control of important cell differentiation proteins becomes completely ineffective (Figure 3C, Figure 5B), thus highlighting the importance proteolysis in complementing gene regulatory mechanisms. We have demonstrated this concept by focusing on the newly identified Lon substrates StaR and FliK. Investigating additional putative Lon substrates to establish a more “global role” for Lon will be an interesting task for the future, but would in our eyes go beyond the scope of this manuscript.